# Bilophila wadsworthia aggravates high fat diet induced metabolic dysfunctions in mice

Jane M. Natividad[1], Bruno Lamas[1,2], Hang Phuong Pham [3], Marie-Laure Michel[1], Dominique Rainteau[2,4], Chantal Bridonneau[1], Gregory da Costa[1], Johan van Hylckama Vlieg[5], Bruno Sovran[1], Celia Chamignon[1], Julien Planchais[1], Mathias L. Richard [1], Philippe Langella[1], Patrick Veiga[5] & Harry Sokol [1,2,4,6]

Dietary lipids favor the growth of the pathobiont Bilophila wadsworthia, but the relevance of this expansion in metabolic syndrome pathogenesis is poorly understood. Here, we showed that B. wadsworthia synergizes with high fat diet (HFD) to promote higher inflammation, intestinal barrier dysfunction and bile acid dysmetabolism, leading to higher glucose dysmetabolism and hepatic steatosis. Host-microbiota transcriptomics analysis reveal pathways, particularly butanoate metabolism, which may underlie the metabolic effects mediated by B. wadsworthia. Pharmacological suppression of B. wadsworthia-associated inflammation demonstrate the bacterium's intrinsic capacity to induce a negative impact on glycemic control and hepatic function. Administration of the probiotic Lactobacillus rhamnosus CNCM I-3690 limits B. wadsworthia-induced immune and metabolic impairment by limiting its expansion, reducing inflammation and reinforcing intestinal barrier. Our results suggest a new avenue for interventions against western diet-driven inflammatory and metabolic diseases.

[1] Micalis Institute, INRA, AgroParisTech, Université Paris–Saclay, 78350 Jouy-en-Josas, France. [2] Sorbonne Universités, UPMC Univ. Paris 06, École normale supérieure, CNRS, INSERM, APHP Laboratoire des Biomolécules (LBM), 27 rue de Chaligny, 75012 Paris, France. [3] ILTOO Pharma, 14 rue des reculettes, 75013 Paris, France. [4] Laboratoire des Biomolécules, Département de chimie, École normale supérieure, UPMC Univ. Paris 06, CNRS, PSL Research University, 75005 Paris, France. [5] Danone Nutricia Research, 91767 Palaiseau, France. [6] Department of Gastroenterology, Saint Antoine Hospital, Assistance Publique—Hopitaux de Paris, UPMC, 75571 Paris, France. Correspondence and requests for materials should be addressed to H.S. (email: harry.sokol@aphp.fr)

In the last three decades, the prevalence of obesity and associated metabolic complications such as type 2 diabetes and non-alcoholic fatty liver disease have significantly increased worldwide and represent a major socio-economic burden[1]. Hence, there is an increasing drive to understand factors that may influence the development and progression of the metabolic disease.

Different diets rapidly and reproducibly alter both the composition and function of the gut microbiota[2]. For instance, over-representation of *Bilophila wadsworthia*, a Gram-negative sulfite-reducing bacteria that is commonly recovered from patients with appendicitis[3,4], had been associated with animal based diets and diets rich in fats[2,5–7]. At least in HFD setting, increased production of taurine conjugated bile acids had been proposed to underlie the expansion of *B. wadsworthia*[5]. The negative effect of increased abundance of *B. wadsworthia* on intestinal inflammation had been demonstrated, but despite its documented association with dietary fats[5,7], it remains unknown whether *B. wadsworthia* imposes negative consequences on metabolic host function. Interestingly, higher level of *B. wadsworthia* had been similarly observed in individuals suffering from severe malnutrition[8]. The mechanism by which a single bacterium can play a pleiotropic role in myriad of diseases remained unclear, but diet seemed to play a major role in driving microbial fitness[9].

It has now been recognized that alterations in gut microbiota composition and function seem to be one hallmark of metabolic impairment;[1] however, causal relationships that underlie these processes are complex and are not yet fully understood. Nevertheless, the ability to access and reprogram the composition and function of the gut microbiota make it an attractive target for preventive or therapeutic intervention. Oral deliveries of specific probiotics strains belonging to *Lactobacillus* and *Bifidobacterium* species have shown to confer protective effects against obesity and metabolic syndrome in animal models[10]. Little is however known whether these beneficial bacteria directly modulate microbiota function and composition, which in turn limit disease progression.

Here, using a combination of host transcriptomics, metatranscriptomics, gnotobiotics and conventional mouse models, we explored the ability of the pathobiont *B. wadsworthia* to thrive under HFD setting (saturated animal-derived fat) and its ability to modify host physiology and metabolism. Furthermore, we investigated the capacity of the probiotic *Lactobacillus rhamnosus* strain CNCM I-3690, previously demonstrated to have an anti-inflammatory properties, protective effects against intestinal barrier dysfunction and HFD-induced metabolic alterations in mice, as well as able to induce a decrease of Desulfobrionaceae family, on which *B. wadsworthia* belongs to[10–12]. We showed that *B. wadsworthia* worsens the detrimental impact of HFD on host metabolism, in inflammation dependent and independent manners, and *L. rhamnosus* CNCM I-3690 was efficient in delimiting some of the *B. wadsworthia*-associated metabolic and immune impairments, suggesting that preventing *B. wadsworthia*'s expansion may be a novel therapeutic strategy in both inflammatory and metabolic diseases.

## Results

**HFD promotes increased *B. wadsworthia* expansion in mice.** To confirm the effect of dietary fat on the host metabolic status, mice were maintained on either control diet (CD) or high fat diet enriched with milk-fat (HFD). In agreement with previous results[5,7], we observed significantly higher fecal *B. wadsworthia* level in HFD-fed mice but the increase was only 3.7-fold higher compared to CD-fed mice after 9 weeks of diet (Fig. 1a). This

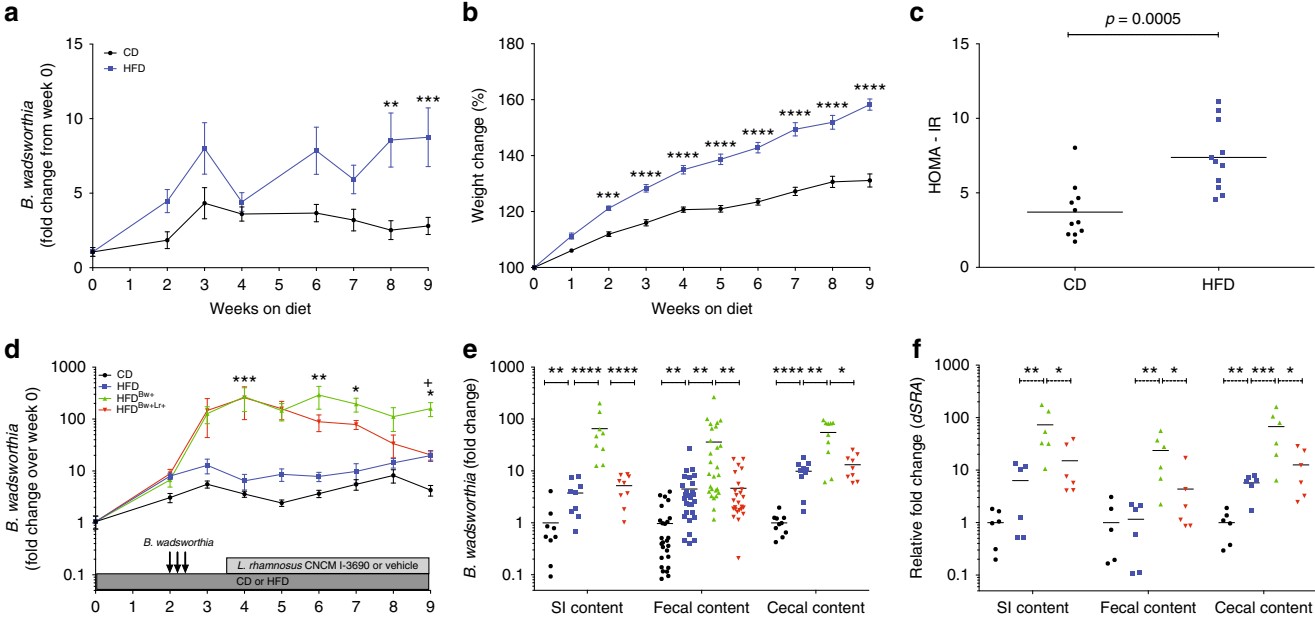

**Fig. 1** *B. wadsworthia* expands in HFD environment. **a** Fold change of *B. wadsworthia* in mice fed with control diet (CD) or high-fat diet (HFD) relative from day 0 (**p < 0.01, ***p < 0.001; n = 11/group). **b** Body weight gain (***p < 0.001, ****p < 0.0001; n = 11/group), **c** Homeostatic model assessment-insulin resistance (HOMA-IR) after 6 h of fasting (n = 10/group). **d** Fold change of *B. wadsworthia* relative from day 0 in mice fed with control diet (CD) or high-fat diet (HFD) and inoculated with *B. wadsworthia* (Bw+) and treated with *L. rhamnosus* CNCM I-3690 (Lr+) (*p < 0.05, **p < 0.01, ***p < 0.001 vs HFD; +p < 0.05 vs HFD^Bw+Lr+; n = 16–28/group). **e** *B. wadsworthia* load in small intestinal (SI), fecal and cecal contents after 9 weeks of CD or HFD. **f** Expression of the *dsra* gene in the small intestinal (SI), fecal and cecal contents after 9 weeks of CD or HFD (*p < 0.05, **p < 0.01, ***p < 0.001, ****p < 0.0001). Statistical comparison was performed by first testing normality using Kolmogorov–Smirnov test, then t-test with Welch correction or Mann–Whitney non-parametric test for two groups, or ANOVA or Kruskal–Wallis test with Bonferroni or Dunn's post hoc test for more than three groups. Error bars represents SEM

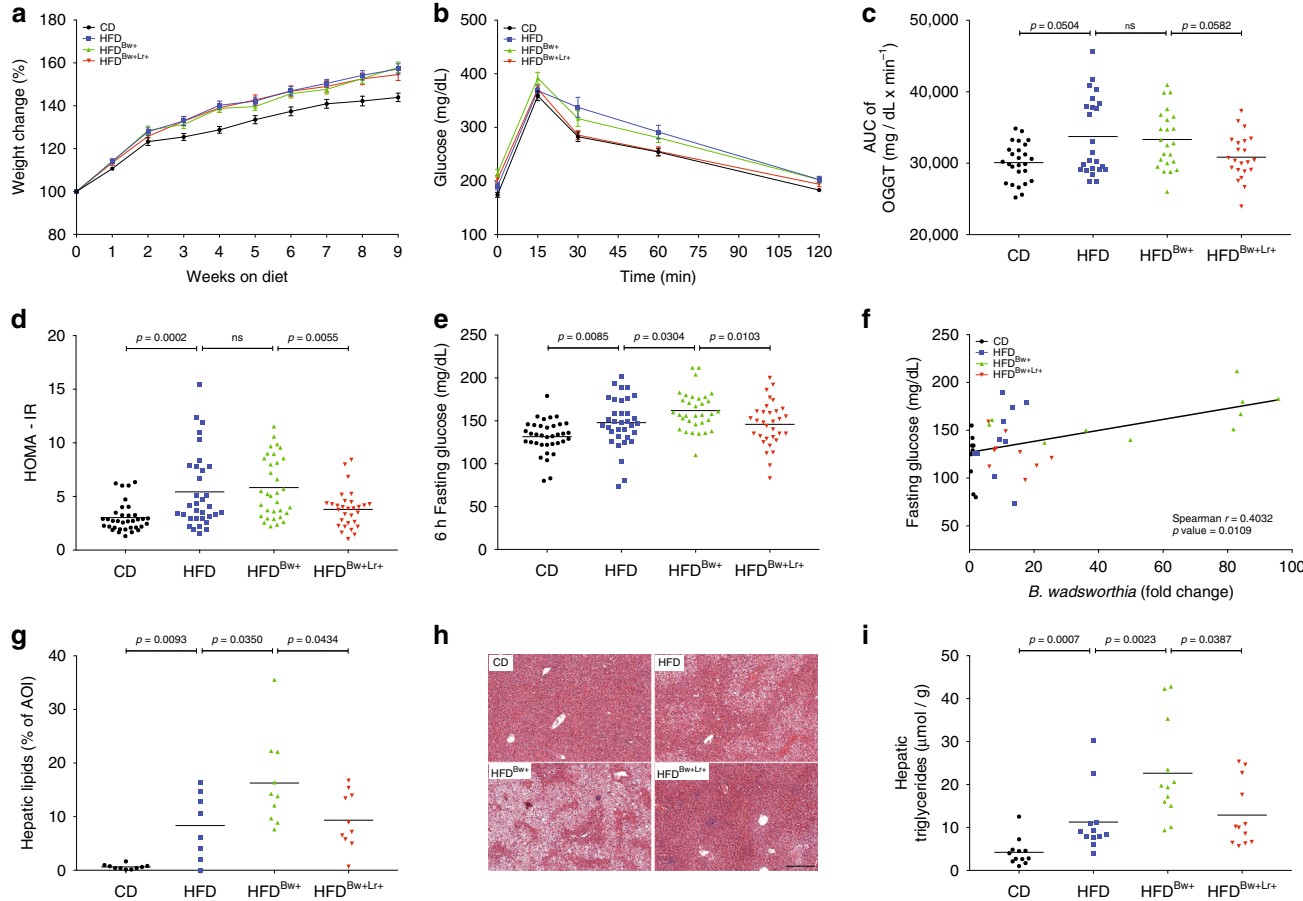

**Fig. 2** *B. wadsworthia* synergizes with HFD to trigger a stronger metabolic impairments. **a** Body weight gain ($n = 37$–40/group). **b** Blood glucose level before and after oral glucose tolerance challenge (OGTT; 2 g/kg mouse; $n = 27$–40/group). **c** Area under the curve (AUC) of OGTT. **d** homeostatic model assessment-insulin resistance (HOMA-IR) after 6 h of fasting. **e** Blood glucose after 6 h of fasting. **f** Spearman correlation of fasting glucose and *B. wadsworthia* load in the cecal content. **g** Lipid area, calculated as % area of interest (AOI), in liver cross-sections stained with H&E. **h** Representative pictures of liver stained with H&E. Scale bar = 100 μm. **i** Liver triglycerides after 6 h of food deprivation. Statistical comparison was performed by first testing normality using Kolmogorov–Smirnov test and then ANOVA or Kruskal–Wallis test with Bonferroni or Dunn's post hoc test. Error bars represents SEM

result might reflect the low level of *B. wadsworthia* in our animal facility. Nonetheless, mice on HFD had gained more than 20% weight compared to CD-fed mice (Fig. 1b). Weight gain was linked to an increased calorie intake (Supplementary Figure 1a). HFD-fed mice further showed elevated fasting blood glucose and insulin as well as homeostatic model assessment–insulin resistance (HOMA-IR) compared to CD-fed controls (Fig. 1c, Supplementary Figure 1b-c). Similarly, blood glucose clearance, evaluated by oral glucose tolerance test (OGTT), was altered, characterized by higher baseline and peak glycaemia and significantly upper area under the curve (AUC), in HFD-fed mice (Supplementary Figure 1d-e). Finally, HFD-fed mice displayed higher hepatic triglycerides compared to CD (Supplementary Figure 1f).

To better determine whether higher density of *B. wadsworthia* affects metabolic functions, *B. wadsworthia* was deliberately given to the mice by intragastric gavage. This protocol induced a higher and stable intestinal level of *B. wadsworthia* (Fig. 1d–f). Interestingly, CD-fed mice gavaged with *B. wadsworthia* similarly showed higher and stable level of *B. wadsworthia* but *B. wadsworthia* fecal abundance was still significantly higher in HFD-fed mice (Supplementary Figure 2). These results underscored that HFD is not necessary for *B. wadsworthia* to thrive in the host intestine; however, it is essential for the sustained higher

levels of *B. wadsworthia*. We thus generated two animal models: (1) mice harboring low *B. wadsworthia* (HFD) and (2) mice harboring high-density levels of *B. wadsworthia* (HFD$^{Bw+}$). Using these models, we were able to discriminate the relevance of *B. wadsworthia* abundance in metabolic host function.

**B. wadsworthia aggravates HFD-induced metabolic impairments.** To determine the consequence of *B. wadsworthia* abundance on host metabolic status, metabolic parameters were evaluated in mice after a period of HFD feeding. We did not observe significant differences in weight gain between HFD mice harboring low or high *B. wadsworthia* abundance (Fig. 2a). Similarly, no significant differences in weekly food intake were observed among HFD-fed groups (Supplementary Figure 3a). Glucose clearance, as assessed by OGTT as well as insulin level and HOMA-IR were not affected by increased *B. wadsworthia* abundance in HFD-fed mice (Fig. 2b–d). However, HFD$^{Bw+}$ mice showed higher fasting glucose compared to HFD group (Fig. 2e). Furthermore, a strong positive correlation between fasting glucose and *B. wadsworthia* load in the caecum was observed (Fig. 2f).

Elevated serum liver enzymes, hepatic steatosis as well as cholesterol levels are commonly observed in obese individuals[13]. Significant increase in serum concentrations of aspartate

transaminase (AST) and alanine transaminase (ALT) were observed in all HFD groups compared to CD-fed mice, but there was no significant difference between HFD and HFD$^{Bw+}$ groups (Supplementary Figure 3c–d). Similarly, all HFD-fed mice, regardless of treatment, have significantly elevated levels of total cholesterol and HDL in the plasma (Supplementary Figure 3e–f). On the other hand, analysis of liver histology revealed that hepatic lipid content was significantly increased in HFD$^{Bw+}$ group (Fig. 2g, h). In parallel, total hepatic triglyceride was significantly higher in HFD$^{Bw+}$ group than HFD (Fig. 2i), showing that *B. wadsworthia* has detrimental effects on this metabolic feature. Taken together, these results showed that the high abundance of *B. wadsworthia* potentiates specific HFD-induced host metabolic syndrome, with notable dysregulation of glucose homeostasis and liver function.

**L. rhamnosus CNCM I-3690 prevents B. wadsworthia expansion**. Having established a robust murine model of HFD-driven metabolic syndrome with stable and high levels of *B. wadsworthia* and the consequence of *B. wadsworthia* abundance on metabolic syndrome pathogenesis, we tested the therapeutic potential of the *L. rhamnosus* CNCM I-3690 strain in this model. Daily oral gavage of *L. rhamnosus* CNCM I-3690 (Lr) induced a significant decrease in fecal *B. wadsworthia* load (Fig. 1a–c). Similarly, *L. rhamnosus* CNCM I-3690 was able to further reduce *B. wadsworthia* expansion in caecum and small intestine.

We then determined whether preventing *B. wadsworthia* expansion by *L. rhamnosus* CNCM I-3690 has host metabolic consequences. *L. rhamnosus* CNCM I-3690 did not affect weight gain and food intake in HFD and in HFD$^{Bw+}$ mice (Fig. 2a; Supplementary Figure 3a). *L. rhamnosus* treated HFD$^{Bw+}$ mice (HFD$^{Bw+Lr+}$) showed reduced fasting glucose level, plasma insulin and HOMA-IR response (Fig. 2d, e, Supplementary Figure 3b). OGTT further revealed that HFD$^{Bw+Lr+}$ mice tended (AUC of OGTT $p = 0.0582$) to control glucose level better than HFD$^{Bw+}$ (Fig. 2b). *L. rhamnosus* CNCM I-3690 did not have any effect on glycemic control in HFD group with lower *B. wadsworthia* level (Supplementary Figure 4a, d–e) but it corrected the effect of HFD on insulin level (Supplementary Figure 4b), suggesting that in addition to suppressing *B. wadsworthia*-related metabolic dysfunctions by preventing its expansion in vivo, it also improves metabolic function by its inherent ability to modulate insulin level.

**HFD feeding modulates microbiota composition**. HFD had been consistently shown to modulate intestinal microbial community; thus, we evaluated the global microbiota changes induced by the HFD diet as well as the microbiota effect of inoculating the mice with *B. wadsworthia* and *L. rhamnosus* CNCM I-3690 using 16s rRNA-based high throughput sequencing technique. Regardless of treatment, fecal microbiota of CD group clustered differently from mice fed with HFD (Fig. 3a), highlighting a dominant effect of diet (Anosim, 9999 permutations, $p = 0.0001$). Compared to CD, HFD-fed mice had lower abundance of bacteria belonging to the genera *Ruminococcus*, *Bifidobacterium* and *Parabacteroides* and of *Akkermansia muciniphila* species (Fig. 3b, c). On the other hand, higher abundance of bacteria under *Dorea* and *Sutterella* genera and *Ruminococcus gnavus* species was observed in HFD-fed mice. Inoculating mice with *B. wadsworthia* did not induce major changes in microbiota composition in HFD-fed mice (Fig. 3a), although an increase in abundance of *Akkermansia* and *Bifidobacterium* genera was observed in HFD$^{Bw+}$ group (Supplementary Figure 5a). Similarly, *L. rhamnosus* CNCM I-3690, did not induce significant changes in microbiota

composition; however, as expected, an increase abundance in an OTU related to *L. rhamnosus* was observed in HFD$^{Bw+Lr+}$ group (Supplementary Figure 5b). No significant differences in alpha diversity were observed among groups as measured by observed species, Chao1 and Shannon indexes (Fig. 3d and Supplementary Figure 5c). Overall, these data showed that HFD has a significant impact on microbiota composition but the effect of *B. wadsworthia* and *L. rhamnosus* CNCM I-3690 on the microbiota composition, assessed by 16S-based approach, was limited and may be more relevant at the metabolic functional level.

**B. wadsworthia modulates host and microbial transcriptomics**. To explore the interplay between *B. wadsworthia*, host metabolism, gut microbiota and *L. rhamnosus* CNCM I-3690, we chose to work in a controlled microbiota environment, that is a gnotobiotic mice colonized with the eight species of the so-called altered Schaedler flora (ASF)[14]. We further exploited this system by conducting a comprehensive transcriptomic analysis on both the host and bacterial genes. Germ-free mice were colonized with ASF, and then maintained on either CD or HFD. ASF-colonized mice were then inoculated with *B. wadsworthia* and treated with (HFD-ASF$^{Bw+Lr+}$) or without *L. rhamnosus* CNCM I-3690 (HFD-ASF$^{Bw+}$). *B. wadsworthia* colonization was confirmed by measuring *B. wadsworthia* density in the caecum (Fig. 4a).

Host (caecum) transcriptomics analysis identified many differentially regulated transcripts in HFD compared to CD fed mice. In HFD-ASF mice, expression of 302 genes was significantly altered compared to CD-ASF mice (Supplementary data 1) whereas this number reached 1630 genes in HFD-ASF$^{Bw+}$ mice demonstrating stronger gene modulation. Compared to HFD-ASF mice, HFD-ASF$^{Bw+}$ exhibited an activation in many pathways involved in inflammation and immune response and a marked alteration in fat and glucose metabolism and regulation. Notably Type II diabetes mellitus and Insulin signaling pathways were respectively over-activated and under-activated (Fig. 4b, Supplementary data 2). These alterations were largely reversed in HFD-ASF$^{Bw+Lr+}$ with correction of most of the altered pathway activation seen in HFD-ASF$^{Bw+}$ mice (Fig. 4b, Supplementary data 2). To confirm these results, we measured cytokines at the protein level in mesenteric lymph nodes (MLN), spleen and caecum and showed that, IFN-γ and IL-6 production were consistently higher in tissues of HFD-ASF$^{Bw+}$ mice compared to other groups and *L. rhamnosus* CNCM I-3690 reversed this phenotype (Fig. 4c).

In parallel, we analyzed in the same animals the functional activity of ASF microbiota, *B. wadsworthia* and *L. rhamnosus* CNCM I-3690 using metatrancriptomics strategy and RNAseq technology (Fig. 5a, and Supplementary data 3). The global microbiota activity of HFD-ASF mice was dysregulated with increased activity in sugar interconversions pathways and decreased activity in nitrogen metabolism pathway. Microbial gene expression in all HFD-fed mice colonized with *B. wadsworthia* showed strong alteration with notably significantly higher activation of Lipopolysaccharide (LPS) biosynthesis and Taurine metabolism pathways while many pathways involving amino acid, sugar, starch, and nitrogen metabolism were significantly reduced. To further evaluate whether the increased LPS production at the microbiota expression level have physiological relevance to the host, we measured serum soluble CD14 (sCD14), a component of the LPS-sensing machinery used as a surrogate marker for LPS concentration in the periphery, and found significantly higher sCD14 in HFD-ASF$^{Bw+}$ mice (Fig. 5b).

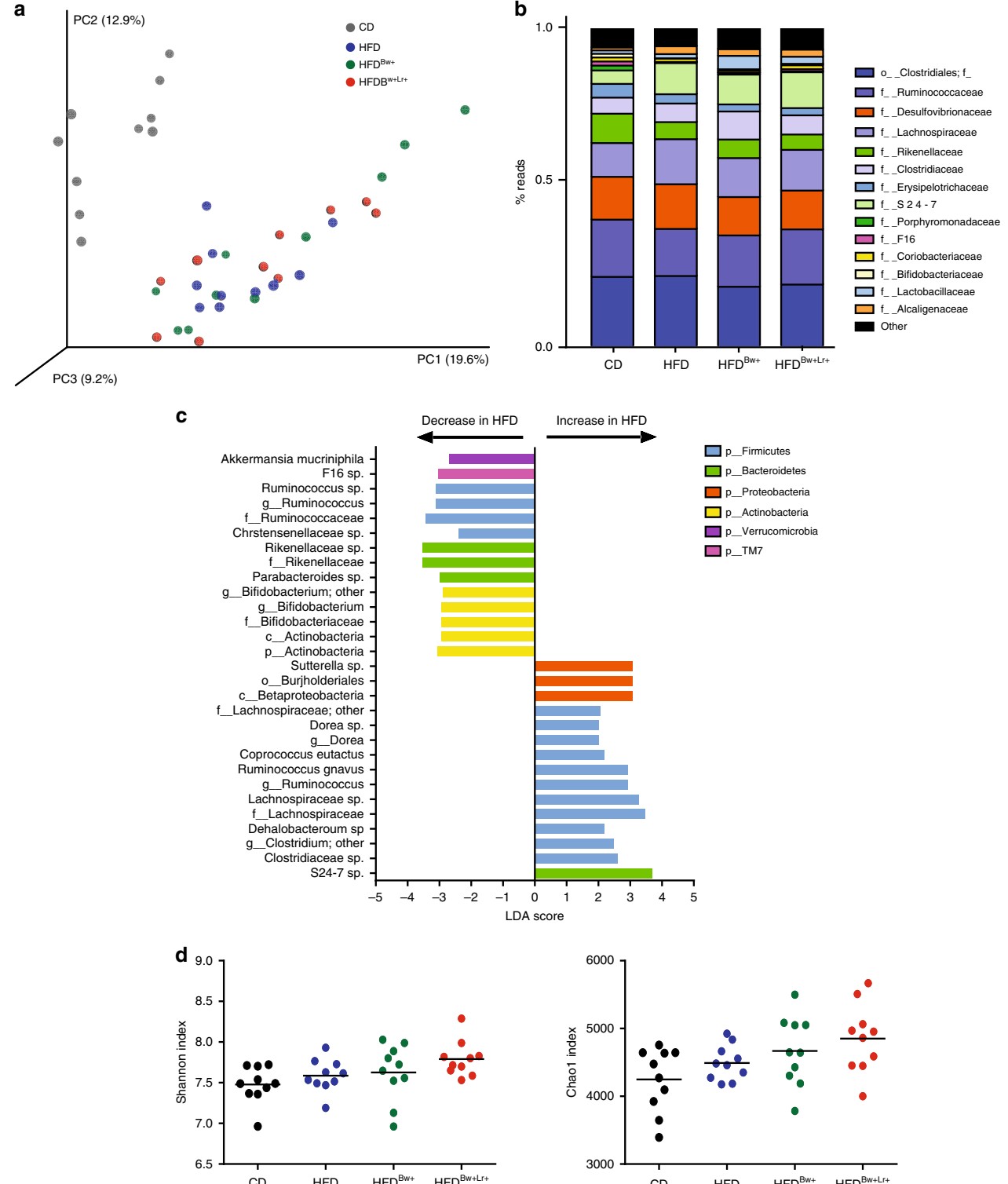

**Fig. 3** HFD induces significant impact on microbiota composition. **a** PCoA plot (Bray Curtis distance) of fecal microbiota of CD-fed or HFD-fed mice inoculated with or without *B. wadsworthia* (Bw) and/or *L. rhamnosus* CNCM I-3690 (Lr) on Bray Curtis distance. **b** Bar graph of bacterial abundance in family level. **c** Bacterial taxa differentially enriched in HFD- compared to CD-fed mice determined using Linear discriminant analysis (LDA) effect size (LEfSe) algorithm. **d** Fecal microbiota alpha diversity. Statistical comparison was performed by first testing normality using Kolmogorov–Smirnov test and then ANOVA or Kruskal–Wallis test with Bonferroni or Dunn's post hoc test. For each group, two cages of five animals were used and analyzed

Finally, compared to other groups, only HFD-ASF^(Bw+) mice showed decreased activation of Butanoate (butyrate) metabolism pathway, an alteration reverted in HFD-ASF^(Bw+) treated with *L. rhamnosus* CNCM I-3690. We further confirmed this last result

by measuring the fecal concentration of SCFA. A strong tendency ($p = 0.0593$) for reduced butyrate concentration was observed in HFD-ASF^(Bw+) compared to HFD-ASF. Moreover, HFD-ASF^(Bw+Lr+) mice showed significantly higher butyrate compared to HFD-

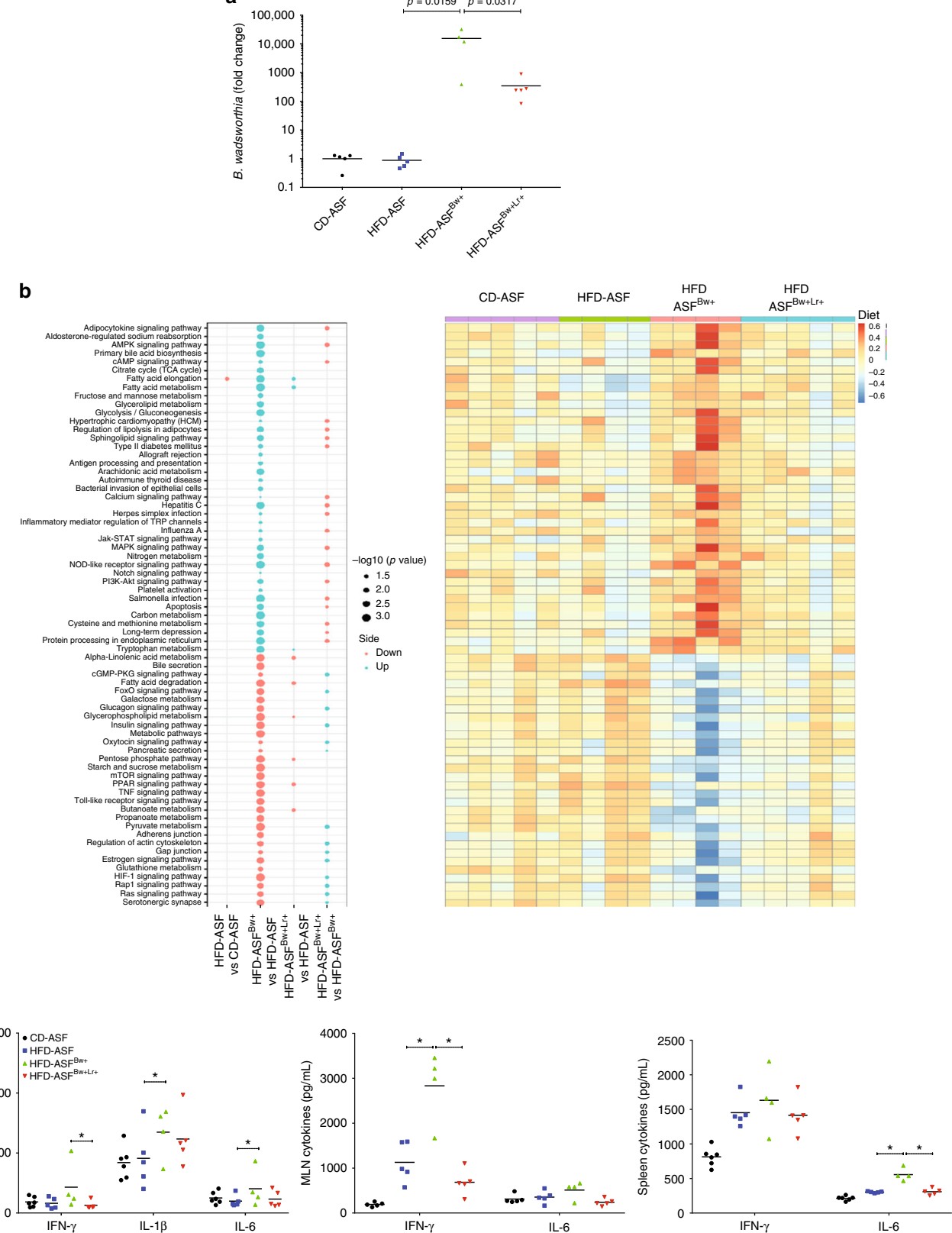

**Fig. 4** *B. wadsworthia* regulates expression of inflammatory and metabolic-related host genes under HFD setting. **a** Fold change of *B. wadsworthia* (Bw) relative from altered Schaedler flora (ASF)-colonized mice fed with control diet (CD). **b** Left: Bubble plot of -log10(BH *p*-values) of a selection of significant pathway activities between indicated groups (Blue: up regulated in first compared to second group. Red: down regulated in first compared to second group); right: Heatmap of corresponding pathway activity. **c** Cytokine production in caecum, mesenteric lymph node and spleen (*$p < 0.05$ vs. HFD-ASF, +$p < 0.05$ vs. HFD-ASF$^{Bw+}$; $n = 4$–5/group). Statistical comparison was performed by first testing normality using Kolmogorov–Smirnov test and then ANOVA or Kruskal–Wallis test with Bonferroni or Dunn's post hoc test. Error bars represents SEM

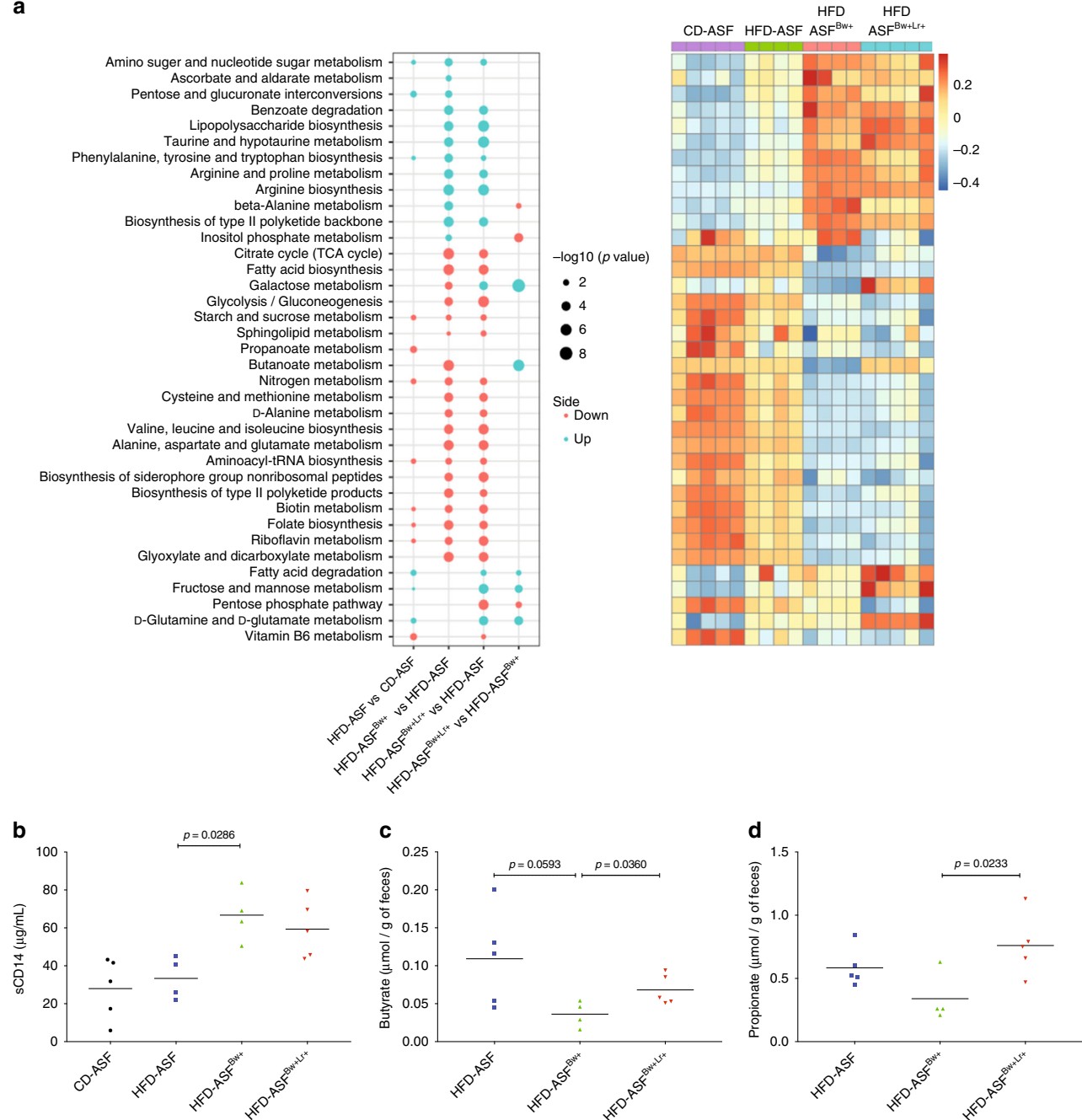

**Fig. 5** The presence of *B. wadsworthia* and *L. rhamnosus* CNCM I-3690 induces changes in microbiota metatranscriptomics. **a** Left: Bubble plot of -log10(BH *p*-values) of a selection of significant pathway activities between indicated groups (Blue: up regulated in first compared to second group. Red: down regulated in first compared to second group); right: Heatmap of corresponding pathway activity. **b** Concentration of soluble CD14 in serum. **c** Butyrate and **d** propionate concentrations in the feces. Statistical comparison was performed by first testing normality using Kolmogorov–Smirnov test and then ANOVA or Kruskal–Wallis test with Bonferroni or Dunn's post hoc test. Error bars represents SEM

ASF$^{Bw+}$ (Fig. 5c). Interestingly, HFD-ASF$^{Bw+}$ also showed lower fecal propionate concentration compared to HFD-ASF$^{Bw+Lr}$ group (Fig. 5d).

Taken together, these results showed that *B. wadsworthia* acts on both host and microbiota by worsening HFD-induced intestinal inflammation, inhibiting pathways involved in metabolic homeostasis, favoring increased LPS production and translocation, and decreasing butyrate production by the microbiota. Most of these *B. wadsworthia*-associated alterations were fully or partly reversed by *L. rhamnosus* CNCM I-3690 administration.

**B. wadsworthia enhances HFD-induced bile acid dysmetabolism.** Our host transcriptomic data revealed that *B. wadsworthia* modulates a number of genes involved in taurine metabolism, which is linked with bile acid homeostasis. Bile acids are increasingly recognized as important signaling factors and regulators of metabolism[15,16]. As such, we investigated the bile acid profile of mice harboring complex microbiota. Indeed, we found that HFD feeding leads to changes in bile acid composition in the caecum; these alterations were characterized by significantly higher total bile acids and elevated primary bile acids conjugates, as opposed to secondary conjugates (Fig. 6a–c). Furthermore,

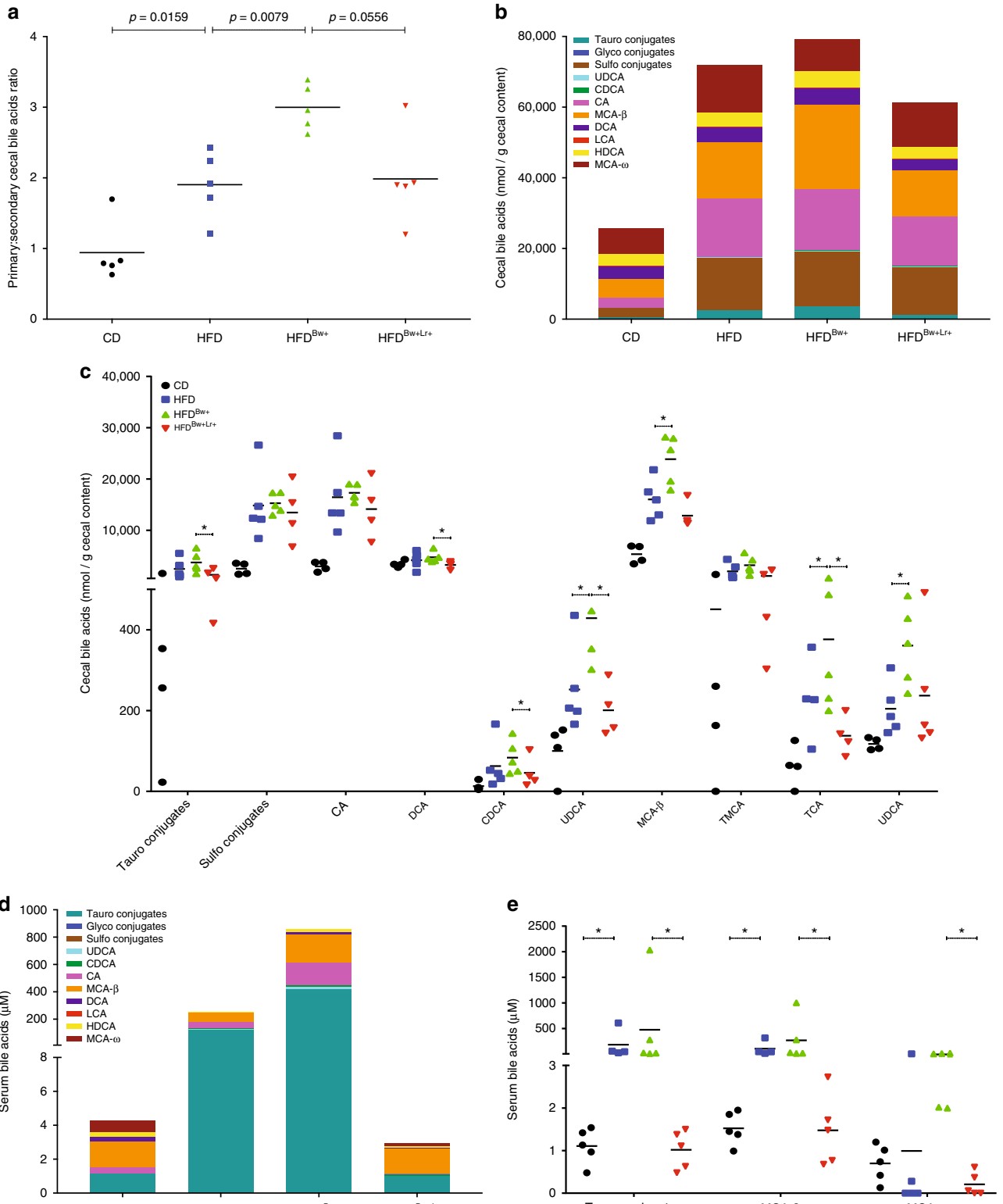

**Fig. 6** *B. wadsworthia* worsens HFD-induced bile acid dysmetabolism. **a** Ratio of primary to secondary bile acids in caecum. **b** Stacked bar showing the bile acids concentration in the caecum. **c** Concentration of difference bile acids in the caecum. (*$p < 0.05$ vs. HFD, +$p < 0.05$ vs. HFD$^{Bw+}$; $n = 5$–6/group). **d** Stacked bar showing the bile acids concentration in the serum. **e** Concentration of difference bile acids in the serum (*$p < 0.05$, **$p < 0.05$; $n = 5$–6/group). Statistical comparison was performed by first testing normality using Kolmogorov–Smirnov test and then ANOVA or Kruskal–Wallis test with Bonferroni or Dunn's post hoc test. Error bars represents SEM

cecal bile acids of mice fed with HFD showed decreased proportion of bile acids such as DCA and HDCA. *B. wadsworthia* tends to further dysregulate bile acid composition in the caecum with higher levels of taurocholic acid (TCA), a taurine-conjugated bile acid, as well as other bile acids such as UDCA and MCA-β. Furthermore, in the serum of HFD-fed mice, taurine conjugated bile acid concentration was more than 100-fold higher compared to CD, with an even stronger increase in HFD$^{Bw+}$ group (Fig. 6d, e). In contrast, HFD$^{Bw+Lr+}$ showed lower total and taurine-conjugated bile acids compared to HFD$^{Bw+}$, suggesting the efficiency of *L. rhamnosus* CNCM I-3690 to reverse the effect of HFD and *B. wadsworthia* on bile acids.

***L. rhamnosus* CNCM I-3690 reverts *B. wadsworthia* host dysfunction**. Based on our simplified microbiota studies, the presence of *B. wadsworthia* downregulated the production of SCFA and upregulated the global synthesis of LPS by the intestinal microbial communities and was further associated with higher systemic LPS. Guided by these results, we similarly assessed the SCFA concentration in caecum and the LPS availability in the systemic compartment in our HFD conventional mice model. In accordance with the results obtained from ASF-colonized mice, butyrate concentration tended to be lower in the caecum of HFD$^{Bw+}$ than in HFD mice and *L. rhamnosus* CNCM I-3690 was associated with a higher level of butyrate and propionate (Fig. 7a, b). Similarly, serum sCD14 level was significantly higher in HFD$^{Bw+}$ than in HFD mice, but *L. rhamnosus* CNCM I-3690 did not reverse this phenotype (Fig. 7c).

Intestinal barrier dysfunction is an important feature in obesity and metabolic syndrome[17]. We hypothesized that this parameter may underlie the increased systemic bioavailability of LPS. Thus, we assessed intestinal permeability using a classical permeability marker FITC-dextran. HFD$^{Bw+}$ mice exhibited increased intestinal permeability as demonstrated by higher serum FITC-dextran levels following oral gavage (Fig. 7d). Intestinal permeability was dampened by *L. rhamnosus* CNCM I-3690 in HFD$^{Bw+}$ but not HFD (Fig. 7d, Supplementary Figure 6a), suggesting this phenotype is modulated by *L. rhamnosus* through regulation of *B. wadsworthia* abundance. Overall, these results show that the increased *B. wadsworthia* abundance augments the impact of HFD-induced gut barrier alterations and *L. rhamnosus* CNCM I-3690 partially reverse this effect.

Disruption of the gut barrier may allow increased intestinal permeability to bacterial endotoxins, such as LPS, and in turn may increase mucosal inflammation and lead to systemic inflammation. Hence, we next examined whether *B. wadsworthia* further exacerbates HFD-induced inflammatory response in conventional mice. We first characterized the state of mucosal inflammation by quantifying lipocalin-2 levels in the feces on different time-points during the experiment (Fig. 7e). HFD feeding tended to show higher levels of lipocalin-2 in the feces compared to CD but this was further and significantly increased in HFD$^{Bw+}$ mice, particularly at week 7 and 9. Cytokine levels in MLN, ileum and jejunum of HFD$^{Bw+}$ were similarly higher compared to either CD or HFD or both groups, underscoring a state of heightened mucosal inflammation in HFD$^{Bw+}$ group (Fig. 7f–h). *L. rhamnosus* CNCM I-3690 treatment was able to dampen some of these responses, particularly for fecal lipocalin-2 levels, TNF-α and IFN-γ in HFD$^{Bw+}$ but not in HFD group (Fig. 7e–h, Supplementary Figure 6b).

We further assessed the state of systemic inflammation and observed a similar pattern with significantly increased production of several pro-inflammatory cytokines such as IFN-γ, TNF-α and IL-6 in the spleen and liver of HFD$^{Bw+}$ mice (Fig. 7i, j). Similar to mucosal immune response, *L. rhamnosus* CNCM I-3690 treated

mice displayed a cytokine pattern closer to the control groups, suggesting its ability to dampen the pro-inflammatory effect of *B. wadsworthia*. All together, these results showed that *B. wadsworthia* synergizes with HFD in inducing higher states of systemic and mucosal inflammation, which can be at least partly reversed by *L. rhamnosus* CNCM I-3690.

***B. wadsworthia's* inflammation-independent modulating properties**. In accordance with previous study[5], we observed that *B. wadsworthia* possessed inherent pro-inflammatory properties; however, it remains unknown at what extent this *B. wadsworthia* feature plays a role in development of host metabolic impairment. Inflammation is an important feature of metabolic syndrome; consequently, it is also unclear whether *B. wadsworthia*-associated metabolic impairments are just secondary to *B. wadsworthia*-driven inflammation. To address these questions, inflammation in HFD-fed mice was suppressed using a broadly used immunosuppressant, ciclosporine. Cytokine analysis in spleen, MLN and intestine as well as Lipocalin-2 level in stools showed that ciclosporine (Ci) effectively abrogated the inflammatory response in HFD-fed mice, regardless whether they harbor low or high-density *B. wadsworthia* (Fig. 8a, b; Supplementary Figure 7a–c). Thus, by normalizing the inflammatory state between the HFD-Ci and HFD-Ci$^{Bw+}$, potential interference of inflammation can be eliminated and direct metabolic effect of *B. wadsworthia* can be clearly inferred.

No difference in *B. wadsworthia* density was observed between ciclosporine treated and untreated HFD$^{Bw+}$, ruling out any direct effect of ciclosporine treatment on *B. wadsworthia* (Fig. 8c). Ciclosporine treated HFD and HFD$^{Bw+}$ mice showed significantly less weight gain compared to non-treated mice, underscoring the importance of inflammation in promoting weight gain (Fig. 8d). Regardless of weight change however, ciclosporine did not affect fasting glucose, insulin and HOMA-IR in HFD-fed mice (Fig. 8e–g). However, HFD$^{Bw+}$ mice treated with ciclosporine showed significantly elevated fasting insulin and this was associated with a strong tendency ($p = 0.0576$) for higher HOMA scores. OGTT further showed that ciclosporine-treated HFD-fed mice tend to exhibit improved glucose tolerance compared to non-treated mice; but HFD$^{Bw+}$ mice still had worse glucose tolerance, as evidenced by higher glycemic level 15 min following oral challenge and significantly higher AUC, compared to HFD mice (Fig. 8h, i).

In terms of hepatic effect, ciclosporine did not seem to significantly dampen hepatic steatosis in both HFD (Fig. 8j, k). Although, there was a tendency ($p = 0.0728$) for HFD-Ci$^{Bw+}$ to have lower steatosis compared to HFD$^{Bw+}$ group. Nonetheless, the difference in steatosis between HFD and HFD$^{Bw+}$ remains significant regardless of ciclosporine treatment. Altogether, these results showed that the pro-inflammatory effects of *B. wadsworthia* partly mask its negative metabolic effects. Moreover, *B. wadsworthia* exhibits negative intrinsic metabolic effect independently of inflammation.

## Discussion
HFD had been consistently associated with increased abundance of *B. wadsworthia*, a bacterium implicated in increased colitis severity of *Il-10*$^{-/-}$ mice[5]. However, the impact of *B. wadsworthia* on non-genetically susceptible host, and whether and how its expansion could promote an impaired metabolic function remains poorly understood. Here, we utilized a hypothesis-driven approach, using a combination of host trancriptomics, meta-trancriptomics and gnotobiotic techniques, to dissect how *B. wadsworthia* is able to modulate host metabolic response to HFD. We then tested our hypotheses in conventional HFD

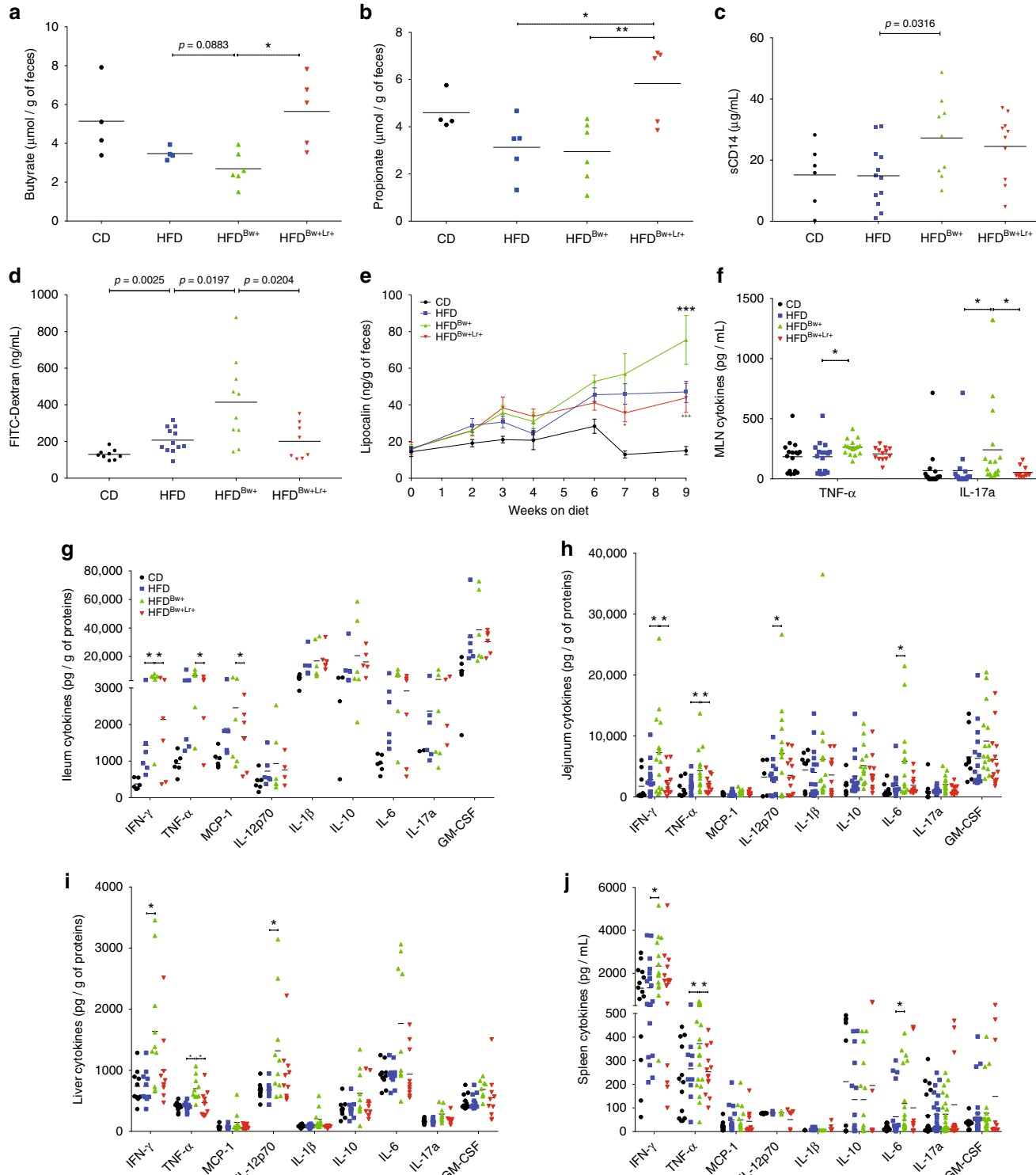

**Fig. 7** *B. wadsworthia* potentiates HFD-induced intestinal barrier dysfunction and inflammation. **a** Butyrate and **b** propionate concentrations in caecum. **c** Soluble CD14 (sCD14) in the serum. **d** Concentration of FITC-dextran in the serum 3 h post-gavage. **e** Concentration of lipocalin-2 in the feces. **f** Cytokine production of mesenteric lymph node (MLN) cells after 48 h stimulation with PMA-ionomycin (*p-value vs. HFD, +p-value vs. HFD^Bw+; n = 6–16/group). Cytokines level in **g** ileal, **h** jejunal, and **i** liver homogenates (*p-value vs. HFD, +p-value vs. HFD^Bw+; n = 6–16/group). **j** Cytokine production of splenic cells after 48 h stimulation with PMA-ionomycin (*p-value vs. HFD, +p-value vs. HFD^Bw+; n = 6–16/group). Statistical comparison was performed by first testing normality using Kolmogorov–Smirnov test and then ANOVA or Kruskal–Wallis test with Bonferroni or Dunn's post hoc test. Error bars represents SEM

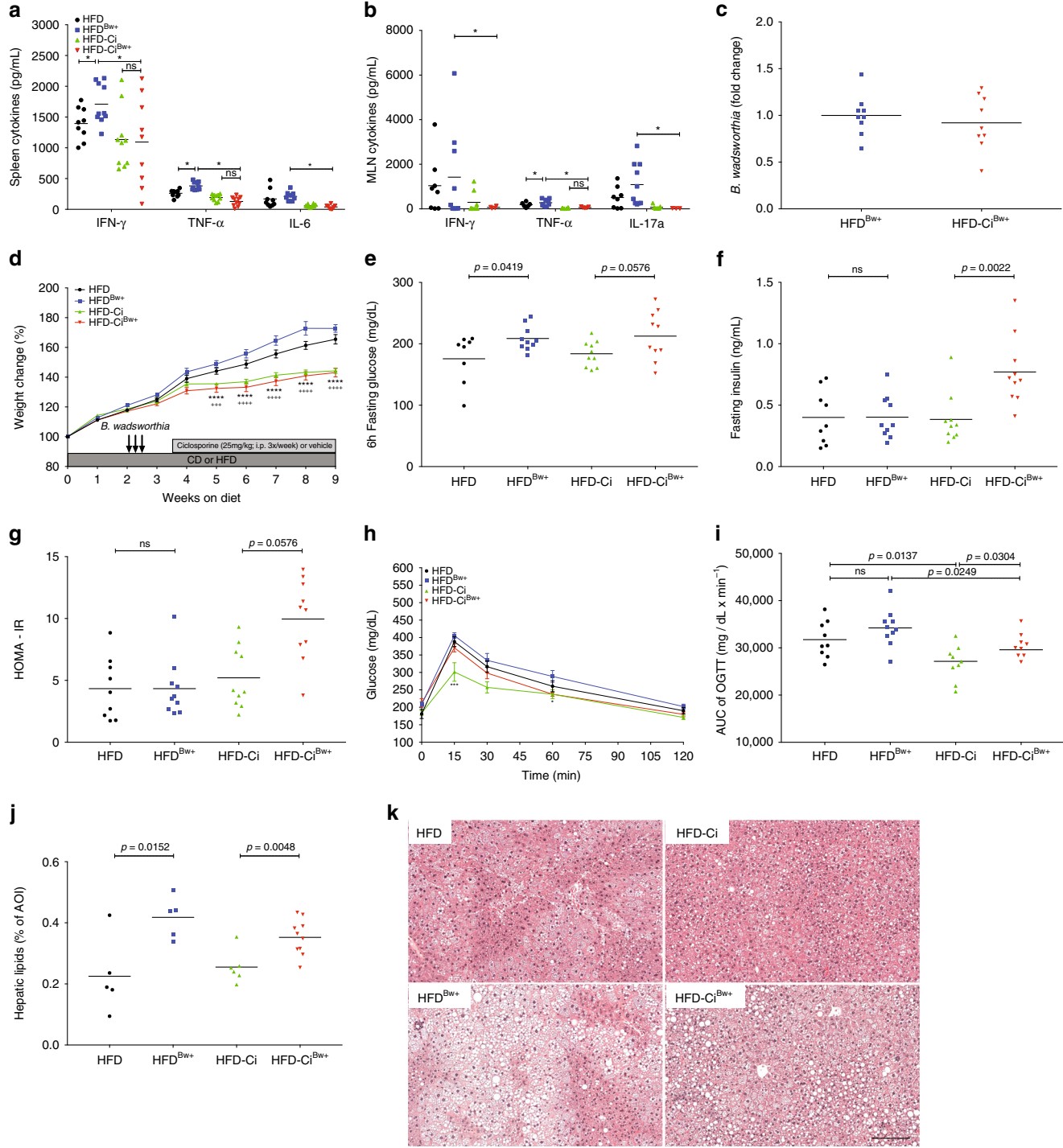

**Fig. 8** Suppression of inflammation unmasks intrinsic effects of *B. wadworthia* on glucose homeostasis. Cytokine production in **a** spleen and **b** MLN of HFD fed mice receiving ciclosporine (Ci) and/or *B. wadsworthia* (Bw+) (*$p < 0.05$, $n = 8–10$/group). **c** *B. wadsworthia* load in the caecum (*$p < 0.05$, $n = 8–10$/group). **d** Weight gain after 9 weeks of HFD (****$p < 0.0001$ HFD vs. HFD-Ci; +++$p < 0.001$, ++++$p < 0.0001$ HFD^Bw+ vs. HFD-Ci^Bw+, $n = 8–10$/group). **e** Blood glucose, **f** insulin, **g** homeostatic model assessment-insulin resistance (HOMA-IR) after 5–6 h of fasting. **h** Blood glucose level before and after oral glucose tolerance challenge (OGTT; 2 g/kg mouse; ***$p < 0.001$, *$p < 0.05$ HFD-Ci vs. HFD-Ci^Bw+; $n = 8–10$/group). **i** Area under the curve (AUC) of OGTT. **j** Lipid area, calculated as % area of interest (AOI), in liver cross-sections stained with H&E. **k** Representative pictures of liver stained with H&E. Scale bar = 100 μm. Statistical comparison was performed by first testing normality using Kolmogorov–Smirnov test and then ANOVA or Kruskal–Wallis test with Bonferroni or Dunn's post hoc test. Error bars represents SEM

murine model. Our results showed that, beside intestinal pro-inflammatory effects, *B. wadsworthia* promotes intestinal barrier defect, systemic inflammation, bile acid dysmetabolism and changes in microbiome functional profile, leading to the worsening of HFD-induced metabolic effects. Moreover, we showed that *L. rhamnosus* CNCM I-3960 was able to inhibit most of the *B. wadsworthia*-driven host metabolic and inflammatory impairments.

To determine the mechanistic basis by which *B. wadsworthia* impacts host metabolism and how *L. rhamnosus* CNCM I-3690 attenuates these effects, we performed transcriptomic analysis on both the host and microbiota. To fully understand the system, we chose to work in a controlled microbiota environment, wherein bacterial and host function can be inferred to a specific microbe or condition. Among the gene pathways that were significantly modulated by the presence of *B. wadsworthia* are those involved in bile acid metabolism. Bile acids are synthesized from cholesterol. In the liver, taurine, along with glycine, are used to conjugate bile acids to produce primary bile acids. Bile acids undergo enterohepatic circulation, which includes circulating in the intestine where primary bile acids are further deconjugated and converted into secondary bile acids by the microbiota. Saturated animal-derived fats had been previously shown to promote the production of taurine conjugated bile acids, such as taurocholic acid (TCA)[2,5]. In this study, we observed that HFD significantly upregulates intestinal genes involved in the metabolism of taurine with increased concentration of taurine conjugated bile acids and decreased proportion of secondary bile acids in the caecum. *B. wadsworthia* further aggravates the bile acid deregulation in HFD context and this can be alleviated by *L. rhamnosus* CNCM I-3690 treatment. This is important as secondary bile acids have an important negative feedback role in decreasing bile acid synthesis;[15] hence, the increased total serum and caecum bile acids in HFD$^{Bw+}$ group may be compounded by the decreased negative feedback signaling due to lower proportion of secondary bile acids. Additionally, unlike conjugated bile acids, unconjugated bile acids, such as cholic acid and chenodeoxycholic acid, are strong agonist for bile acid receptors, including Farnesoid X receptor and transmembrane G protein-coupled receptor;[16] signaling through these receptors activates transcriptional networks and signaling cascades relevant for cholesterol and lipid metabolism, maintenance of glucose and hepatic homeostasis, as well as genes involved in suppressing inflammation and strengthening intestinal barrier function[15,18]. Moreover, the pro-inflammatory properties of primary bile acids had been described[19,20]. Altogether, these data suggest that *B. wadsworthia*'s impact on bile acid metabolism may be one mechanism by which the bacterium potentiates HFD-induced metabolic impairment and host dysfunctions, particularly inflammation and barrier dysfunction.

How the presence of *B. wadsworthia* leads to a disproportionate level of bile acids remains an open question, but it is possible that *B. wadsworthia* has an inherent ability to drive the production of primary acid, particularly TCA, for its own advantage. Indeed, it had been previously shown that *B. wadsworthia* expands in the presence of taurine conjugated bile acids, especially TCA[5]. Similarly, we observed that *B. wadsworthia* grow in vitro ten times more in the presence of 1% taurine (Supplementary Figure 8). Thus, in conjunction with previous results, this suggests that taurine and its derivatives, particularly TCA, may not be necessary for *B. wadsworthia*'s survival but are essential for its increased fitness and growth.

Our metatranscriptomics studies revealed also that LPS synthesis pathway was highly upregulated in HFD$^{Bw+}$ mice microbiota. This was paralleled by higher LPS translocation, which may at least partly explain the increased systemic inflammatory response we observed in HFD$^{Bw+}$, both in ASF-colonized and conventional mice. *L. rhamnosus* CNCM I-3690 may be reducing the pro-inflammatory phenotype in HFD$^{Bw+}$ mice by decreasing the abundance of *B. wadsworthia*, and also by decreasing the bioavailability of LPS in the systemic circulation and/or through its intrinsic anti-inflammatory effects that had been previously demonstrated in other murine models[11]. Although *L. rhamnosus* CNCM I-3690 was efficient in decreasing

inflammatory markers and intestinal permeability in HFD$^{Bw+}$, it was ineffective in decreasing these phenotypes in HFD mice, highlighting that *L. rhamnosus* CNCM I-3690 acts through *B. wadsworthia* modulation in this setting.

In addition to LPS synthesis, the presence of *B. wadsworthia* induced a decreased expression of microbial genes involved in butanoate metabolism in ASF-colonized mice. Furthermore, the decreased production of butyrate was confirmed by dosage in colon lumen of the mice. Aside from its effect in modulating inflammatory response[21], butyrate had been shown enhance intestinal epithelial barrier by assembly of tight junctions[22,23]. Furthermore, dietary supplementation with butyrate had been previously shown to confer preventive and therapeutic benefits in animal model of obesity and insulin resistance[24]. Interestingly, *L. rhamnosus* strains had been shown to be efficient in compensating butyrate deficiency and enhance intestinal barrier[11,25], thereby providing another potential mechanism of action of *L. rhamnosus*.

We showed that *B. wadsworthia* increased HFD-induced metabolic syndrome, which is a condition often associated with low degree of systemic inflammation[26]. At the same time, *B. wadsworthia* had been demonstrated to exacerbate intestinal inflammation in genetically susceptible mice[5] and also to induce systemic inflammation in wild type animals[27]. Higher state of inflammation is characterized by wasting and weight loss while metabolic syndrome is characterized by low grade inflammation and higher body mass index. Thus, a key question is how can *B. wadworthia* affect both opposing pathologies. Moreover, the contribution and relative importance of *B. wadworthia*'s pro-inflammatory properties in disturbing host metabolic status in HFD setting is unknown. By pharmacologically suppressing the inflammation, we unmasked the capacity of *B. wadsworthia* to directly induce a negative impact on host metabolic function. Specifically, we were able to discriminate distinct metabolic impairments, namely reduced glucose clearance and fatty liver phenotype, influenced by *B. wadworthia* that do not completely rely on its pro-inflammatory properties. Nonetheless, *B. wadworthia*-driven inflammation remains an important factor that further tip the balance to stronger metabolic dysfunctions. Accordingly, this may explain why *B. wadworthia* is able to play a pathological role on two contrasting settings. It is also important to note that although we showed in this study that *B. wadsworthia* promoted heightened immune response, this did not translate to patent colitis development compared to previous reports[5]. This might be due to the fact that our murine model is not a genetically susceptible host, and therefore have intact compensatory mechanism that prevents the development of overt intestinal inflammation. Hence, *B. wadsworthia*'s effect on metabolic function in our system outweighs its effect on development of overt inflammation due to lack of additional stimuli. As a result, the phenotype we achieved is a model by which *B. wadsworthia* potentiates the deleterious effect of HFD on host metabolic homeostasis. Taken all together, the effect of *B. wadsworthia* to the host can be pleiotropic, but the combination of genetics, host and environmental factors ultimately dictate the degree of inflammation and type of pathologies *B. wadworthia* will exacerbate or influence.

Overall, we demonstrated that *B. wadworthia* augments some of the deleterious host metabolic effects of HFD by modifying multiple targets: altering the functional potential of intestinal microbes, promoting higher intestinal permeability, development of heightened immune response both at mucosal and systemic level, and disproportionate bile acid concentrations. All these pathways ultimately converge to further disturb the host metabolic function in HFD setting (Supplementary Figure 8). Importantly, we also unraveled that *B. wadsworthia*, independent

of its pro-inflammatory properties, has the capacity to negatively affect glucose and hepatic homeostasis. These results suggest that the carriage of high levels of *Bilophila* species or strains might modulate the susceptibility to not only inflammatory but also metabolic diseases. Collectively, our study provides a conceptual framework to further test this hypothesis in human and warrants the evaluation of preventative strategies, such as probiotics use, to suppress the expansion of *Bilophila*.

## Methods

**Experimental model**. For conventional experiment, male C57BL/6J mice were purchased from Janvier (France) and used after 1 week of receipt. Mice at 5 weeks of age were fed *ad libitum* with purified control diet (CD, Envigo TD.120508) or high fat diet (HFD, 38% fat, dominantly milk fat, Envigo TD.97222) for 9 weeks. For deliberate *B. wadsworthia* inoculation, after maintaining the mice in HFD or CD for 2 weeks, mice were inoculated via oral gavage with ~$10^7$ CFU of *B. wadsworthia* ATCC 49260 suspended in 200 µl of medium (Bacteroides bile esculin with 1% Taurine and 0.5 mg/ml cysteine) or medium alone for three consecutive days. For *L. rhamnosus* CNCM I-3690 treatment, 1 week after the last *B. wadsworthia* inoculation, mice were gavaged daily with $10^9$ CFU of *L. rhamnosus* CNCM I-3690 suspended in 200 µl of vehicle (phosphate buffer saline with 15% glycerol) or vehicle for 5 weeks. For ciclosporine experiment, 1 week after the last *B. wadsworthia* inoculation, mice were injected i.p. with ciclosporine (25 mg/kg; Sandimmum Novartis) or vehicle (PBS) 3x a week for 5 weeks.

For altered Schaedler flora (ASF) experiment, male C57BL/6J germ-free (GF) mice were obtained from Transgenese et Archivage Animaux Modeles (CNRS, UPS44, Orleans, France) and used after 1 week of receipt. Sterility was confirmed microscopically and by microbiological technique. ASF colonized mice were kindly provided by E. Verdu from McMaster University (Canada). Fresh cecal samples from ASF-colonized mice were suspended and diluted in pre-reduced sterile 0.9% NaCl with 15% glycerol (1 g in 10 ml) under anaerobic condition. Aliquots of ASF cecal suspension were stored at −80 °C. GF mice (5 weeks of age) were inoculated via oral gavage with 200 µl of ASF cecal suspension and maintained on either HFD or CD. 3 weeks after ASF inoculation, mice were orally gavaged with *B. wadsworthia* or medium for 3 consecutive days. 1 week after the last *B. wadsworthia* inoculation, mice were gavaged daily with $10^9$ CFU of *L. rhamnosus* CNCM I-3690 or vehicle for 4 weeks. ASF mice were housed in isolator for the whole experiment.

Weekly food consumption was measured cage-wise. Mice were fasted for 6 h prior to sacrifice and then put to sleep using isoflurane. Mice were culled by cervical dislocation and appropriate tissues were harvested. All experiments were performed in accordance with the French animal experimental committee (Comite d'Ethique en Experimentation Animale).

**Oral glucose tolerance test**. Oral glucose tolerance test was performed 3–5 days before the sacrifice. Mice were fasted by removing the food and bedding 1 h before the onset of light cycle. After 6 h of fasting, glucose solution (2 g/kg) was administered by oral gavage. Blood glucose level at time 0 (fasting glucose, taken before glucose gavage) and at 15, 30, 60, and 120 min after glucose gavage was analyzed using OneTouch glucometer (Roche). Glucose level was plotted against time and areas under the glucose curve (AUC) were calculated by following trapezoidal rule. Plasma insulin concentration (collected in EDTA-coated tubes) at time 0 (fasting insulin) and 30 was analyzed from tail vein blood (collected in EDTA-coated tubes) using ultra sensitive mouse insulin ELISA kit (Alpco). Homeostatic model assessment of insulin resistance (HOMA-IR) was calculated according to the formula: fasting glucose (nmol/L) x fasting insulin (microU/L)/22.5.

**Measurements of plasma parameters**. Blood samples were collected in heparin-coated tubes via cardiac puncture, centrifuged and then plasma samples were stored at −80 °C. Plasma cholesterol, triglycerides, high-density lipoprotein (HDL), aspartate transaminase (AST) and alanine transaminase (ALT) measurement were performed by the Biochemistry Platform (CRI, UMR 1149, Paris) using Olympus AU400 Chemistry Analyzer.

**Measurements of bile acids**. Measurement of bile acids (BA) composition and concentration in plasma and intestinal contents was performed by the Chemistry department at Saint Antoine Hospital (UMR 7203, France) using high performance liquid chromatography (HPLC, Agilent 1100, France) coupled in series with mass spectrometer (QTRAP 2000, Canada)[20].

**Measurements of SCFA**. Measurement of the short-chain fatty acids (SCFA) from fecal content was performed by the mass spectrometer platform at Universite de Nantes (IRS-UN, France) using gas chromatography coupled with mass spectrometry[28].

**Quantification of cytokines**. Single cell suspensions from MLN and spleen were isolated by smashing the cells in 70 µm mesh. $1 \times 10^6$ cells were plated in 24 well-plate and then stimulated with phorbol 12-myristate 13-acetate (PMA, 50 ng/mL; Sigma-Aldrich) and ionomycin (1 µM; Sigma Aldrich) for 48 h at 37 °C. Supernatants were collected and used for cytokine analysis.

Fifty milligram of intestinal tissues and liver samples were suspended in T-PER Tissue Protein Extraction Reagent (Thermo Scientific) and homogenized suing FastPrep (6 m/s in 40 s). Homogenates were centrifuged and supernatants were used for cytokine and total protein concentration analysis. Total protein concentration of the tissue homogenates were analyzed using Pierce BCA Protein Assay Kit (Thermo Scientific). Cytokine concentrations were normalized according to the measured protein concentration.

Cytokines were measured using Legendplex Mouse Inflammation Panel (Biolegend) or individual ELISA kit (R&D Mouse DuoSet IL-6; Mabtech IFN-γ, IL-17a ELISA kits; Ebioscience TNF-α ELISA kit).

**Liver histology and hepatic triglycerides measurement**. A slice of left lobe of the liver was fixed in 4% PFA for 48 h and then transferred to ethanol, fixed in paraffin, trimmed, processed, sectioned into slices approximately 3 µm thick, mounted on a glass slide and stained with hematoxylin and eosin (H&E). Hepatic lipids were evaluated by quantifying the % lipid area using the software ImageJ[29].

**In vivo intestinal permeability and plasma sCD14**. In vivo assay of intestinal barrier function was performed using fluorescein-conjugated dextran (FITC-dextran, 3–5kDA) method, as previously described[30]. Briefly, on the day of sacrifice, FITC-dextran (0.6 mg/g of body weight) was administered to the mice by oral gavage and 3 h later, blood samples were collected in heparin-coated tubes. Fluorescence intensity was measured in the plasma using a microplate reader (Tecan). Plasma concentration of soluble CD14 (sCD14) was measured using CD14 ELISA kit (R&D).

**Quantification of fecal LCN2**. Frozen fecal samples were weighed and reconstituted in cold PBS. Samples were then agitated on a FastPrep bead beater machine for 40 s at setting 6 using 4.5 mm glass beads to obtain homogenous fecal suspension. Samples were then centrifuged for 5 min at 10,000×*g* (4 °C) and clear supernatants were collected and stored at −20 °C until analysis. LCN2 levels were estimated using Duoset murine LCN2 Elisa Kit (R&D) as per manufacturer's instructions and expressed as pg/mg of stool.

**DNA extraction and bacterial quantification**. Fecal, cecal and small intestinal content genomic DNA was extracted from the weighted stool samples using a method that was previously described[31], which is based on the Godon DNA extraction method. Quantifications of all bacteria and *B. wadsworthia* were performed by qPCR using TaqMan Gene Expression Assays (Life technologies) and Takyon SYBR Green PCR kit (Eurogentec). All bacteria was quantified using the following oligonucleotides: (sense) 5′-CGGTGAATACGTTCCCGG-3′ and (anti-sense) 5′-TACGGCTACCTTGTTACGACTT-3′ and (probe) 5′-CTTGTACACAC CGCCCGTC-3′. *B. wadsworthia* was quantified using specific primers for the *tpa* gene (accession no. AF269146): (sense) 5′-CGCCGGTATCGAAATCGTGA-3′ and (antisense) 5′-ATTCGCGGAAGGAGCGAGAG-3′. Sulfite-reducing bacteria were quantified using specific primers for the *dsra* gene (encoding a dissimilatory sulfite reductase alpha subunit) as described by Devkota[5].

**16s rRNA gene sequencing**. 16s rRNA gene sequencing of fecal DNA samples (collected at week 9 of CD or HFD) was performed as previously described[31]. Briefly, the V3-V4 region was amplified and sequencing was done using an Illumina MiSeq platform (Illumina). Sequencing data was analyzed using the quantitative insights into microbial ecology (QIIME 1.9.1) software package. Sequences were clustered into operational taxonomic unites (OTU) at a 97% identity threshold using a closed-reference picking approach with UCLUST against the Greengenes reference database (version 13.5) and phylogenetic tree was built using FastTree. Rarefaction was performed (13,000 sequences per sample) and used to compare abundance of OTUs across samples. Alpha-diversity was estimated using both richness and evenness indexes (Chao1, Shannon or number of observed species). Beta-diversity was measured Bray Curtis distance matrix and was used to build principal coordinates analysis (PCoA) plots. Linear discriminant analysis (LDA) effect size (LEfSe) algorithm was used to identify taxa that are specific to diet and/or treatment. Sequencing data are deposited in European Nucleotide Archive (ENA) under the accession number PRJEB25364.

**Mouse gene expression and microarray analysis**. Total RNA was isolated from the caecum of ASF-colonized mice using RNeasy Mini Kit, according to manufacturer's instructions. The RNA integrity was verified in Bioanalyzer 2100 with RNA 6000 Nano chips (Agilent Technologies). Only samples that have RNA integrity >9 were used for the study. Mouse transcriptomics was performed using SurePrint G3 Mouse GE 8x60K Microarray (Design ID: 028005, Agilent Technologies), according to manufacturer's instructions. Microarray data are

deposited in GEO under the accession number GSE111451 [https://www-ncbi-nlm-nih-gov.gate2.inist.fr/geo/query/acc.cgi?acc=GSE111451].

Microarray data processing and analysis was performed as previously described[28]. Briefly, Agilent Feature Extraction Software v10.7.3.1 was used to convert scanned signal into tab-delimited text files. p-value for each probe in each array was computed to test whether the scanned signal is significantly higher than background signal. Probes were filtered according to two criteria: only probes with p-value lower than 0.05 were considered and present in at least 60% of arrays in at last one biological groups. Signals were then log2 transformed, normalized according to *quantiles* method, corrected for batch effect using ComBat method[32] with covariate and Agilent's probes ID were mapped to EntrezGeneID for further analysis. Differential analysis was performed using empirical Bayesian test (eBayes). Beside of analyzing of significant differentially expressed genes, we applied the functional class scoring (FCS) method "Pathway Level Analysis of Gene Expression (PLAGE, embedded in the GSVA R package)"[33] to quantify the level of activity of each KEGG pathway in each sample and utilized empirical Bayes test to compare for the differentially enrichment between biological groups. Significant terms were selected at Benjamini-Hochberg's method corrected p-value lower than 0.05.

**Microbial gene expression and RNA-sequencing analysis**. Total bacterial RNA from cecal content was extracted by mechanic bead-beating lysis method combined with phenol/choloroform RNA extraction method. Briefly, cecal content was suspended in 200 μl of RNAse free water and then 250 μl phenol (pH 4.8)/chloroform-isoamylalcohol mixture (5:1 ratio), 12.5 μl of SDS (20%), 25 μl NaAc (3 M, pH 8) were added to the suspension. Cecal content suspension was then agitated on a FastPrep bead beater machine for 40 s at setting 5 followed by 60 s in ice for another round of bead beating for 20 s at setting 5. After lysis, the upper phase of the suspension was collected after 15 min of centrifugation at 13,000×g (4 °C). Traces of phenol were eliminated by adding 250 μl of chloroforme-isoamylalchohol and then collecting the upper phase after 10 min of centrifugation. Bacterial RNA was then purified using High Pure RNA Isolation Kit (Roche), as per manufacturer's instructions. Total RNA concentration and integrity was determined using a Bioanalyzer 2100 (Agilent).

Library preparation and RNA-sequencing was performed by the High-throughput Sequencing Platform of I2BC. Total bacterial RNA were purified using RiboZero. Library was sequenced using NextSeq 500 on the single lane 75 bp paired-end mode.

Metatranscriptomics data processing and analysis was performed by first trimming unmapped reads and then were aligned using Bowtie2 aligner[34] to genome sequences retrieved form NCBI RefSeq database except for *L. rhamnosus* CNCM I-3690, which was mapped using a genome generated from Danone Nutricia. Mapped reads was quality filtered and only reads with MAPQ ≥ 5 was used. Reads were counted against transcript features database (General Feature Format (GFF) files from NCBI RefSeq database except for *L. rhamnosus* CNCM I-3690 which was predicted using Prokka v1.12b) using the mode "union". Count data was further filtered using the following criteria: genes with 0 read across all samples and genes with <1 count per million were excluded. For visualization, count data was transformed using Variance Modeling at the Observation Level (voom method)[35] and then normalized by quantile method. Differential analysis was performed using the exact tests for differences between two groups of negative binomial counts programmed in the R package "edgeR"[36]. All the significant signatures (up-regulated or down-regulated genes) were annotated for enriched pathways using the package GOstats[37]. Pathway definition for bacteria was downloaded from PATRIC database. Pathway lists were cleaned to exclude host pathways that were predicted from the same enzyme commission numbers (EC) assigned to both bacteria and host genes. As for host microarray, the method PLAGE was used to compare the pathway activity level between groups. p-values were adjusted using Benjamini-Hochberg's method to correct for multiple-testing problem. Significant threshold is fixed at alpha = 0.05 (type 1 error). All metatranscriptomics were deposited on GEO repository under the accession number GSE112387 [https://www-ncbi-nlm-nih-gov.gate2.inist.fr/geo/query/acc.cgi?acc=GSE112387].

**Statistical analysis**. In each experiment, multiple mice were analyze as biological replicates. No statistical methods were used to predetermine sample size. Sample size was estimated according to previous experience using the models described. No samples, mice or data points were excluded from the reported analysis. Animals were randomly assigned to each experimental groups. All analyses were performed unblinded except the histological analyses. Dot plots with a linear scale show the arithmetic mean. Bar graphs are expressed as mean ± standard error of mean (SEM). Except for 16s rRNA, microarray and RNA-sequencing results, GraphPad Prism version 7.0b was used for all statistical analysis. The Kolmogorov–Smirnov test was used to verify that all data set were normally distributed. For data sets that failed normality, nonparametric tests were used to analyze statistical differences. For comparisons between two groups, significance was determined using two-tailed Student's t-test or nonparametric Mann–Whitney test. For comparisons among more than two groups, one way analysis of variance (ANOVA) followed by post-hoc Bonferroni test or nonparametric Kruskal-Wallis test followed by post hoc Dunn's test and two-way ANOVA corrected for multiple comparison with a Bonferroni test were used. An F or Bartlett's test was performed to determine difference in variances for t-tests and ANOVAs, respectively. An unpaired Student's t-test with Welch's correction was applied when variances were not equal. Differences were noted as significant at p ≤ 0.05.

**Data availability**. Sequencing data are deposited in European Nucleotide Archive (ENA) under the accession number PRJEB25364. Microarray data are deposited in GEO under the accession number GSE111451. All metatranscriptomics were deposited on GEO repository under the accession number GSE112387. All other data are available from authors.

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

## Acknowledgements

We thank the members of the INRA-Jouy-en-Josas animal facility and the ANAXEM germ-free platform for their assistance with the animal experiments; Dr. Elena Verdu for the ASF samples; Dr. Jean-Marc Chatel for his assistance during the experimental design process with ciclosporine experiments; Julie Riviere and Abdel for their assistance with the histological work, Marjorie Lebarbenchon for technical help. Our work has benefited from the facilities and expertize of the @BRIDGE Histological and Microarray Platform of UMR 1313 GABI, High-throughput Sequencing Platform of I2BC, Biochemistry Platform of UMR 1149 Inflammation Research Center, and IRS-UN Mass Spectrometry Platform of CRNH. J.M.N. holds a fellowship from Canadian Association of Gastro-enterology and Canadian Institute of Health Research. This work was supported by a grant in aid from Danone Nutricia Research.

## Author contributions

Conceptualization, J.M.N., J.v.H.V., P.L., P.V., and H.S; Methodology, J.M.N., P.L., P.V., and H.S; Investigation, J.M.N., B.L., H.P.P., M.L.M., D.R., C.B., G.d.C., B.S., C.C., J.P., M.L.R., P.L., P.V., and H.S; Writing—Original Draft, J.M.N., and H.S; Writing—Review & Editing, J.M.N., M.L.M., M.L.R., P.L., P.V., and H.S; Funding Acquisition, P.L. and H.S; Supervision, H.S.

## Additional information

**Competing interests:** P.V. is a Danone Employee. The remaining authors declare no competing interests.

