## [Peer Review File · Nature Communications]

Reviewers' comments:

Reviewer #1 (Remarks to the Author):

The manuscript by Natividad et al. aimed to study the effect of *Bilophila wadsworthia* on metabolic and inflammatory phenotypes in a high fat diet (HFD) mouse model. The increase in inflammation by this bacterium has been previously reported in another model (colitis) but metabolic dysregulation is a novel finding. Additionally, the authors show that a probiotic strain of *Lactobacillus rhamnosus* (Lr) directly limited the amount of Bw to the levels detected in HFD mice. It is important that a probiotic microbe can outcompete Bw. However, it is not very surprising then that the metabolic dysregulation induced by Bw was ameliorated by Lr. Further, an attempt was made to check if Bw metabolic effects were independent on inflammation but this yielded confusing results as described below.

1. It should be stressed out from the beginning of Results that the fat in diet was the milk fat because most studies employ lard fat-based diets and Bw does not bloom there. The metabolic results reported in suppl. Fig. 3 should be shown in main figs (fig 1 or 2) as these are important. For example, weight gain and high ALT, AST, cholesterol, HDL were not ameliorated by Lr and OGTT was not worse in HFD/Bw mice.
2. What is known about Lr "inherent ability to modulate insulin level" besides results in this study? and why was this strain chosen in the first place? Can other lactobacilli inhibit the growth of Bw?
3. The mechanistic studies with Lr are not very relevant, since alleviation of metabolic disease and other changes in Bw/Lr mice can be explained by the fact that these mice have low levels of Bw similarly to HFD mice.
4. What is reported in Fig. 3c? Legends need to be improved.
5. Host transcriptomic analysis needs to be clarified (fig.4b): are these ~302 genes mentioned in the text? What criteria were used to select these pathways? Some names repeated several times (e.g. several of interleukin-1). Also, several immune genes are actually downregulated in HFD/ASF/Bw mice (top 20-25 pathways) contradicting other results in the manuscript.
6. Figure 5a shows changes in microbial gene expression. However, some of the pathways shown by the heatmap, like "T cell receptor signaling pathway", are only possessed by the host.
7. The transcriptional host and microbiome changes identified in ASF/HFD/Bw colonized mice are not necessarily applicable to HFD/Bw. In fact, Bw shows much higher levels in ASF/HFD/Bw than in HFD/Bw. Some shown host genes have different behavior between the two types of experiments. Microbial gene expression will also likely be different in ASF/HFD/Bw and HFD/Bw mice.
8. The model using cyclosporine was not thoroughly evaluated because the authors only show that inflammatory responses in spleen and MLN being suppressed. What about the intestine? Whether cyclosporine abrogated inflammation in the gut needs to be validated.
9. In Figure 8f, 8g, the levels of fasting insulin and HOMA-IR in HFD-CiBw+ group are even higher than those in the HFDBw+ group despite the low weight gain. In the presence of Bw, instead of improving insulin resistance, inhibition of inflammation with cyclosporine worsened the phenotype. This all is confusing and needs to be further addressed.

Minor:

1. It is a little strange to start off with the results shown in supplement. figures only. Should consider moving some to the main text.
2. In Figure 1c, what is "dSRA"? The author should mention about it;
3. Typing error: oral glucose tolerance test OGTT and not OGGT;
4. Line 115, "reduced", "lower", just seems backwards;
5. Line 118, "Figure 11", "l" should be lower case;
6. Use correct spelling for cyclosporine, it should be with 'y'
7. A recent paper showed systemic inflammation and weight loss in mice after administration of Bw in contrast to this study (PMID: 29090023). This deserves a comment.

Reviewer #2 (Remarks to the Author):

In this manuscript by Jane Natividad and collaborators, the authors investigated the impact of the bacteria *Bilophila wadsworthia* and *Lactobacillus rhamnosus* on high-fat diet regimen. Using complementary approaches, they demonstrated that *Bilophila wadsworthia*, in the context of HFD, increase intestinal inflammation and alter host metabolism. Interestingly, those effects can be partially prevented by *Lactobacillus rhamnosus*, and detrimental impact of *Bilophila wadsworthia* on the metabolism seems to have inflammation-independent component.

Major points:

- "we observed significantly higher fecal *B. wadsworthia* level in HFD-fed mice but the increase was only 3.7 fold higher compared to CD-fed mice after 9 weeks of diet". To me, even with those data, there is not a strong evidence that *Bilophila wadsworthia* is present at low level in those animals. Indeed, all the data reported regarding *Bilophila wadsworthia* quantification in CD and HFD-mice can be explained by the absence of such bacterium in their animal facility. The approach used by the authors is qPCR, which is such a sensitive technique that amplification may happen even if the specific target is absent. Further highlighting my point is the finding, by the authors, of *Bilophila wadsworthia* in ASF mice (figure 4A). Hence, the authors should soften their conclusion and suggest this possibility.
- Did the authors look at *Bilophila wadsworthia* abundance using their 16S data? Moreover, the authors can also use their data figure 4a for CD-ASF and HFD-ASF groups: are the cycle threshold obtained significantly higher than in CD-conventional and HFD-conventional? Again, I think that this is an important point here, that can elegantly demonstrate that a single bacterium can explain differences observed between vivarium.
- Running the comparison between HFD and HFD+Bw is an elegant approach used by the authors.
- Figure S1C: food intake data should be represented as kcal / day. Saying "weight gain was independent of increased food intake" is not accurate when comparing weight intake between 2 diets containing dramatically different amounts of calories. Same for figure S3b.
- The data presented figure 6D are impressive. *Lactobacillus rhamnosus* is sufficient to reduce serum bile acid levels to a level similar to CD animals. Impressive.
- Regarding the use of cyclosporine in abrogating inflammation, the authors quantified MLNs and spleen cytokines. It is important here to investigate intestinal inflammation, using qPCR or Lipocalin-2 approach.
- Lines 221-222: "Compared to HFD-ASF mice, HFD-ASFBW+ mice showed significantly stronger gene modulation": such statement should be illustrated by PCoA approach.

Minor comments:

- Line 58, please do not use "huge".
- Figure 3A: 2 cages of 5 animals each? Such informations are important and should be reported.
- Figure 3A: please specify that this is using Bray Curtis distance.
- Sequences deposited in public databases should be references with their respective numbers.
- It is unclear, figure 4b, how normalization was performed.
- It is unclear how ASF animals were housed after colonization.
- Lipocalin = Lipocalin-2

Point by point responses to Reviewers

Article: NCOMMS-17-27383

Title: *Bilophila wadsworthia* worsens high-fat diet-induced metabolic impairments in inflammation dependent and independent manners

Reviewer #1:

The manuscript by Natividad et al. aimed to study the effect of *Bilophila wadsworthia* on metabolic and inflammatory phenotypes in a high fat diet (HFD) mouse model. The increase in inflammation by this bacterium has been previously reported in another model (colitis) but metabolic dysregulation is a novel finding. Additionally, the authors show that a probiotic strain of *Lactobacillus rhamnosus* (Lr) directly limited the amount of Bw to the levels detected in HFD mice. It is important that a probiotic microbe can outcompete Bw. However, it is not very surprising then that the metabolic dysregulation induced by Bw was ameliorated by Lr. Further, an attempt was made to check if Bw metabolic effects were independent on inflammation but this yielded confusing results as described below.

Specifics:

1. It should be stressed out from the beginning of Results that the fat in diet was the milk fat because most studies employ lard fat-based diets and Bw does not bloom there. The metabolic results reported in suppl. Fig. 3 should be shown in main figs (fig 1 or 2) as these are important. For example, weight gain and high ALT, AST, cholesterol, HDL were not ameliorated by Lr and OGTT was not worse in HFD/Bw mice.

We have changed the sentence in the beginning of the results to highlight that the fat in our diet was derived from milk as suggested by Reviewer 1, as follows: “Mice were maintained with either on control diet (CD) or high fat diet enriched with milk-fat (HFD).”

Furthermore, we agree that certain data in Supplementary Figure 3 are important findings. As such, we have incorporated some of the results in Supplementary Figure 3 into the main Figure, as suggested by Reviewer 1.

2. What is known about Lr “inherent ability to modulate insulin level” besides results in this study? and why was this strain chosen in the first place? Can other lactobacilli inhibit the growth of Bw?

As stated in the introduction, *L. rhamnosus* CNCM 3690 has been previously demonstrated to have anti-inflammatory properties as well as protective effect against intestinal barrier dysfunction and HFD-induced metabolic alterations in mice. One supplementary and important

reason for this choice is that this strain also induces, in HFD setting, a decrease of Desulfobraceae family to which belongs *B. wadsworthia* (Wang et al.2015 ISME J). This has now been added in the introduction. Interestingly, this effect was not observed with another *Lactobacillus* (*L. paracasei* CNCM I-4270) suggesting some strain-related specificity (Wang et al.2015 ISME J).

[Redacted]

[Redacted]

3. The mechanistic studies with Lr are not very relevant, since alleviation of metabolic disease and other changes in Bw/Lr mice can be explained by the fact that these mice have low levels of Bw similarly to HFD mice.

We agree with the Reviewer's 1 comment that the metabolic alleviation by *L. rhamnosus* in HFD^{Bw} group may be due to the lower abundance of *B. wadsworthia*. Nonetheless, we humbly disagree regarding the relevance of the mechanistic studies and think that it remains valid. Indeed, one the plausible mechanism action by which *L. rhamnosus* alleviates metabolic impairments is the inhibition of *B. wadsworthia*'s abundance.

4. What is reported in Fig. 3c? Legends need to be improved.

Figure 3c reports the bacterial taxa differentially enriched in HFD- compared to CD-fed mice determined using Linear discriminant analysis (LDA) effect size (LEfSe) algorithm. We have now improved the legends in Figure 3c, as suggested by the Reviewer 1.

5. Host transcriptomic analysis needs to be clarified (fig.4b): are these ~302 genes mentioned in the text? What criteria were used to select these pathways? Some names repeated several times

(e.g. several of interleukin-1). Also, several immune genes are actually downregulated in HFD/ASF/Bw mice (top 20-25 pathways) contradicting other results in the manuscript.

The host and microbiota transcriptomics data have now been fully reanalyzed as detailed in the revised Material and Method section of the manuscript. New figures were built to better illustrate the results.

Briefly, functional class scoring is a set of pathway analysis tools for analyzing Omic data. The main goal of pathway analysis methods focuses on the detecting of changes in expression of specific predefined sets of genes that are known in advance (e.g. Gene Ontology, KEGG pathways). We used the method “Pathway level analysis of gene expression using singular value decomposition” (Tomfohr et al., BMC Bioinformatics 2005, 6: 225) to quantify activities of pathways in a sample. Briefly, for each list of genes involved in a same pathway, expression value matrix was extracted from the filtered log-transformed and quantiles normalized expression data set. Gene expression matrix was standardized to have zero mean and unit variance. Singular value decomposition of this expression matrix allows to define the pathway activity level in each sample which is the coefficient of the first right-singular vector of the decomposition. This coefficient is a weighted sum of standardized expression of each gene involved in the pathway. The weight for each gene comes from the left-singular vector of the decomposition. A positive activity level indicates the high proportion of genes with positive weight and high standardized expression. This process was repeated for all pathways. The heatmap shows activity level computed for all KEGG pathways. After computation of pathway activity level, we applied empirical Bayesian test to compare the difference between mouse groups. Significant pathways were selected at Benjamini-Hochberg adjusted p-values less than 0.05. Beside the heatmap of pathway activity level, we represented the $-\log_{10}(\text{BH p-value})$ as bubble plot for significant pathways in each comparison.

For metatranscriptomic data, count data were first filtered and transformed by voom method then normalized by the *quantiles* method. PLAGS method was applied as for the host transcriptomic data describe above.

6. Figure 5a shows changes in microbial gene expression. However, some of the pathways shown by the heatmap, like “T cell receptor signaling pathway”, are only possessed by the host.

We thank the reviewer for this comment. This mistake is due to bad annotations in databases. As mentioned above, the host and microbiota transcriptomics data have now been fully reanalyzed as detailed in the revised Material and Method section of the manuscript.

7. The transcriptional host and microbiome changes identified in ASF/HFD/Bw colonized mice are not necessarily applicable to HFD/Bw. In fact, Bw shows much higher levels in ASF/HFD/Bw than in HFD/Bw. Some shown host genes have different behavior between the two types of experiments. Microbial gene expression will also likely be different in ASF/HFD/Bw and HFD/Bw mice.

We agree with this Reviewer 1 that certain host and microbiota transcriptomic findings may not be necessarily similar in the ASF colonized and conventional mice. The idea of using the ASF colonized mice is to use it as a hypothesis-driven approach, and findings from this study were only used as a framework to further dissect how *B. wadsworthia* is able to modulate host metabolic responses to HFD and not to draw definitive conclusion.

Knowing the caveats imposed by the ASF studies, this is reason why key findings we observed based on the transcriptomics study was confirmed in the conventional model. For instance, inflammation as well as bile acid dysmetabolism were all confirmed in the conventional model. Additionally, we also confirmed that in the conventional HFD model, an alteration in butyrate production was observed in HFD^{Bw} and was corrected by *L. rhamnosus*; this results have now been appended (Figure 7a-b) in the revised manuscript.

8. The model using cyclosporine was not thoroughly evaluated because the authors only show that inflammatory responses in spleen and MLN being suppressed. What about the intestine? Whether cyclosporine abrogated inflammation in the gut needs to be validated.

We agree with the reviewer's comment about inflammation validation in the gut. We have now evaluated the inflammation in the gut after cyclosporine treatment. Fecal Lipocalin-2 level was lower in cyclosporine treated groups compared to non-treated ones. Moreover, Lipocalin-2 level was identical in HFD + Ci and HFD^{Bw+} + Ci groups. Levels of TNF- α and IFN- γ in jejunum and IFN- γ in ileum were decreased in all cyclosporine treated groups. These new findings have now been added in the revised manuscript (Supplementary Figure 8c-e).

9. In Figure 8f, 8g, the levels of fasting insulin and HOMA-IR in HFD-CiBw+ group are even higher than those in the HFDBw+ group despite the low weight gain. In the presence of Bw, instead of improving insulin resistance, inhibition of inflammation with cyclosporine worsened the phenotype. This all is confusing and needs to be further addressed.

Bw worsens metabolic syndrome in HFD-fed mice but it also induces a strong inflammation in the gut. As this strong inflammation might interfere with the intrinsic metabolic effect, we tested the effect of *Bw* after suppressing it using cyclosporin. Indeed, we showed that suppressing the pro-inflammatory effects of *Bw* exacerbated the intrinsic negative metabolic effect of *Bw*. This suggest that the strong pro-inflammatory effect of *Bw* partly inhibit its negative metabolic effect (see model below). This has been clarified in the manuscript.

Minor:

1. It is a little strange to start off with the results shown in suppl. figures only. Should consider moving some to the main text.

We agree with the Reviewer. We have now moved some results to Figure 1.

2. In Figure 1c, what is “dSRA”? The author should mention about it;

Dsra is a bacterial gene encoding a dissimilatory sulfite reductase alpha subunit and is used as a target to quantify Sulphite-reducing bacteria such as *Bw*. This information has now been added in the material and method section.

3. Typing error: oral glucose tolerance test OGTT and not OGGT;

We would like to thank the Reviewer 1 for this typo, we have now corrected this in the manuscript.

4. Line 115, “reduced”, “lower”, just seems backwards;

We would like to thank the Reviewer 1 for this typo, we have now corrected this in the manuscript.

5. Line 118, “Figure 1I”, “I” should be lower case;

We would like to thank the Reviewer 1 for this typo, we have now corrected this in the manuscript.

6. Use correct spelling for cyclosporine, it should be with ‘y’

Indeed, ciclosporine can be spelled in different way including ciclosporin, cyclosporine and cyclosporine.

7. A recent paper showed systemic inflammation and weight loss in mice after administration of Bw in contrast to this study (PMID: 29090023). This deserves a comment.

We would like to thank the Reviewer for pointing this paper. We have read the paper. Feng et al did not use a high fat diet and may explain the discrepancy in some of the results. In our system, we did not see any significant difference in weight gain between HFD and HFD^{Bw}. Nonetheless, similar to ours, they also showed systemic inflammation, which is reassuring. This goes back to one of our conclusion that *B. wadsworthia* is a pleiotropic bacterium and depending on the system, it can behaves differently. This reference has been added in the discussion.

Reviewer #2:

In this manuscript by Jane Natividad and collaborators, the authors investigated the impact of the bacteria *Bilophila wadsworthia* and *Lactobacillus rhamnosus* on high-fat diet regimen. Using complementary approaches, they demonstrated that *Bilophila wadsworthia*, in the context of HFD, increase intestinal inflammation and alter host metabolism. Interestingly, those effects can be partially prevented by *Lactobacillus rhamnosus*, and detrimental impact of *Bilophila wadsworthia* on the metabolism seems to have inflammation-independent component.

Major points:

- "we observed significantly higher fecal *B. wadsworthia* level in HFD-fed mice but the increase was only 3.7 fold higher compared to CD-fed mice after 9 weeks of diet". To me, even with those data, there is not a strong evidence that *Bilophila wadsworthia* is present at low level in those animals. Indeed, all the data reported regarding *Bilophila wadsworthia* quantification in CD and HFD-mice can be explained by the absence of such bacterium in their animal facility. The approach used by the authors is qPCR, which is such a sensitive technique that amplification may happen even if the specific target is absent. Further highlighting my point is the finding, by

the authors, of *Bilophila wadsworthia* in ASF mice (figure 4A). Hence, the authors should soften their conclusion and suggest this possibility.

We would like to thank the Reviewer for this comment. We have now added a sentence regarding the possibility that *B. wadsworthia* is present at a low level in our animal facility.

- Did the authors look at *Bilophila wadsworthia* abundance using their 16S data? Moreover, the authors can also use their data figure 4a for CD-ASF and HFD-ASF groups: are the cycle threshold obtained significantly higher than in CD-conventional and HFD-conventional? Again, I think that this is an important point here, that can elegantly demonstrate that a single bacterium can explain differences observed between vivarium.

We looked at the 16s data which did not show significant difference between groups. This results might be related to the low absolute abundance of Bw despite important fold change when analyzed with qPCR which is a more sensitive and accurate quantification method. The threshold cycle is significantly higher in ASF compared to SPF mice. This may be due to the fact that in ASF, *B. wadsworthia* take advantage of open niches to proliferate compared to what can be seen in SPF mice.

- Running the comparison between HFD and HFD+Bw is an elegant approach used by the authors.

We would like to thank the author for this comment.

- Figure S1C: food intake data should be represented as kcal / day. Saying “weight gain was independent of increased food intake” is not accurate when comparing weight intake between 2 diets containing dramatically different amounts of calories. Same for figure S3b.

We agree with the reviewer. We now show only groups fed with HFD, showing that the amount of calories intake is the same.

- The data presented figure 6D are impressive. *Lactobacillus rhamnosus* is sufficient to reduce serum bile acid levels to a level similar to CD animals. Impressive.

We would like to thank the author for this comment.

- Regarding the use of cyclosporine in abrogating inflammation, the authors quantified MLNs and spleen cytokines. It is important here to investigate intestinal inflammation, using qPCR or Lipocalin-2 approach.

Similar to Reviewer 1, we agree with the reviewer’s comment about inflammation validation in the gut. We have now evaluated the inflammation in the gut after cyclosporine treatment. Fecal Lipocalin-2 level was lower in cyclosporine treated groups compared to non-treated ones.

Moreover, Lipocalin-2 level was identical in HFD + Ci and HFD^{Bw+} + Ci groups. Levels of TNF-a and IFN-y in jejunum and IFN-y in ileum were decreased in all cyclosporine treated groups. These new findings have now been added in the revised manuscript (Supplementary Figure 8c-e).

- Lines 221-222: “Compared to HFD-ASF mice, HFD-ASFBW+ mice showed significantly stronger gene modulation”: such statement should be illustrated by PCoA approach.

The requested PCoA is shown below.

This has not been included in the revised manuscript but, we now clearly indicated that: “compared to CD: 1630 genes with statistically significant altered gene expression in HFD-ASFBw+ mice vs 302 in HFD-ASF mice.”

Minor comments:

- Line 58, please do not use “huge”.

We have now corrected this in the revised manuscript.

.

- Figure 3A: 2 cages of 5 animals each? Such informations are important and should be reported.

We have now corrected this in the revised manuscript.

- Figure 3A: please specify that this is using Bray Curtis distance.

We have now corrected this in the revised manuscript.

- Sequences deposited in public databases should be references with their respective numbers.

We have now corrected this in the revised manuscript.

- It is unclear, figure 4b, how normalization was performed?

The host and microbiota transcriptomics data have now been fully reanalyzed as detailed in the revised Material and Method section of the manuscript. New figures were built to better illustrate the results.

Briefly, functional class scoring is a set of pathway analysis tools for analyzing Omic data. The main goal of pathway analysis methods focuses on the detecting of changes in expression of specific predefined sets of genes that are known in advance (e.g. Gene Ontology, KEGG pathways). We used the method “Pathway level analysis of gene expression using singular value decomposition” (Tomfohr et al., BMC Bioinformatics 2005, 6: 225) to quantify activities of pathways in a sample. Briefly, for each list of genes involved in a same pathway, expression value matrix was extracted from the filtered log-transformed and quantiles normalized expression data set. Gene expression matrix was standardized to have zero mean and unit variance. Singular value decomposition of this expression matrix allows to define the pathway activity level in each sample which is the coefficient of the first right-singular vector of the decomposition. This coefficient is a weighted sum of standardized expression of each gene involved in the pathway. The weight for each gene comes from the left-singular vector of the decomposition. A positive activity level indicates the high proportion of genes with positive weight and high standardized expression. This process was repeated for all pathways. The heatmap shows activity level computed for all KEGG pathways. After computation of pathway activity level, we applied empirical Bayesian test to compare the difference between mouse groups. Significant pathways were selected at Benjamini-Hochberg adjusted p-values less than 0.05. Beside the heatmap of pathway activity level, we represented the $-\log_{10}(\text{BH p-value})$ as bubble plot for significant pathways in each comparison.

For metatranscriptomic data, count data were first filtered and transformed by voom method then normalized by the *quantiles* method. PLAGE method was applied as for the host transcriptomic data describe above.

- It is unclear how ASF animals were housed after colonization.

ASF mice were housed in isolator for the whole experiment. This information ohas been added in the manuscript.

- Lipocalin = Lipocalin-2

We have now corrected this in the revised manuscript.

Reviewers' comments:

Reviewer #2 (Remarks to the Author):

Previous questions have been appropriately addressed.

The only new question is how the PLAGE analysis was used since the original program is not available at the site indicated in Tomfohr et al., 2005.

Reviewer #3 (Remarks to the Author):

I carefully read this revised version, and I am unfortunately not satisfied by the answers the authors gave to my concerns, nor by their incorporated edits. As I stated in my 1st comments, I still think that the most reasonable explanation in their findings is that their animals are initially devoid of *Bilophila*. There is NO data suggesting that *Bilophila* level is low. To me, all the data, including 16S data, point to a full absence. The very modest increase in *Bilophila* following HFD treatment can be easily explain by the diet switch, which will impact bacterial load, DNA extract and qPCR efficiency.

Pretty hard to understand how *Bilophila* load will be different in group of animals fed the same diet under the same conditions (when comparing HFD and HFD + *Bilophila*). I certainly understand that gavaging *Bilophila* will increase the rate of its bloom in HFD treated animals, but the final level should be very similar between HFD and HFD + *Bilophila* IF the bacteria is, as suggested by the authors, initially present in all animals. In other words, gavaging the bacteria should accelerate the bloom, not the final level.

I suggested to compare *B. wadsworthia* relative value of group HFD Figure 1 and group HFD-ASF Figure4a (WITHOUT *Bilophila* gavage), as a nice way of validating their qPCR measurements and investigate if *Bilophila* is indeed absent. This wasn't performed.

If what I said is proved to be correct by the authors, it will make the manuscript even stronger, with the finding that 1 bacterium can, in the context of a HFD, be sufficient to worsen metabolic alterations.

Point by point responses to Reviewers

Article: NCOMMS-17-27383

Title: *Bilophila wadsworthia* worsens high-fat diet-induced metabolic impairments in inflammation dependent and independent manners

Reviewer #2 (Remarks to the Author):

Previous questions have been appropriately addressed. The only new question is how the PLAGE analysis was used since the original program is not available at the site indicated in Tomfohr et al., 2005.

We agree with the reviewer.

PLAGE is embedded within the GSVa R package that can be downloaded from Bioconductor. This has been added in the revised manuscript with the appropriate reference (Hanzelmann J *et al.* GSVa: gene set variation analysis for microarray and RNA-Seq data. BMC Bioinformatics 14, 7 (2013)).

Reviewer #3 (Remarks to the Author):

I carefully read this revised version, and I am unfortunately not satisfied by the answers the authors gave to my concerns, nor by their incorporated edits. As I stated in my 1st comments, I still think that the most reasonable explanation in their findings is that their animals are initially devoid of *Bilophila*. There is NO data suggesting that *Bilophila* level is low. To me, all the data, including 16S data, point to a full absence. The very modest increase in *Bilophila* following HFD treatment can be easily explain by the diet switch, which will impact bacterial load, DNA extract and qPCR efficiency.

Pretty hard to understand how *Bilophila* load will be different in group of animals fed the same diet under the same conditions (when comparing HFD and HFD + *Bilophila*). I certainly understand that gavaging *Bilophila* will increase the rate of its bloom in HFD treated animals, but the final level should be very similar between HFD and HFD + *Bilophila* IF the bacteria is, as suggested by the authors, initially present in all animals. In other words, gavaging the bacteria should accelerate the bloom, not the final level.

I suggested to compare *B. wadsworthia* relative value of group HFD Figure 1 and group HFD-ASF Figure4a (WITHOUT *Bilophila* gavage), as a nice way of validating their qPCR measurements and investigate if *Bilophila* is indeed absent. This wasn't performed.

If what I said is proved to be correct by the authors, it will make the manuscript even stronger, with the finding that 1 bacterium can, in the context of a HFD, be sufficient to worsen metabolic alterations.

We understand the reviewer concern and agree with his suggestion. We thus compared the

abundance of fecal *B. wadsworthia* between the different groups of mice investigated within our study and took as reference SPF mice fed with conventional diet (CD). As shown in the figure below, the abundance of *B. wadsworthia* is 100-fold higher in CD-fed SPF mice than in CD-fed or HFD-fed ASF mice.

In ASF mice, qPCR results indicate that we are at the limit of detection allowed by this technique. Moreover, the fact that *B. wadsworthia* level does not change in HFD-fed ASF mice confirm that there is probably no viable *B. wadsworthia* in ASF mice. Based on these results, we thus consider that the current wording in the manuscript is correct as we mentioned that:
“This result might reflect the low level of *B. wadsworthia* in our animal facility.”